# Liquid-microjet photoelectron spectroscopy of the green fluorescent protein chromophore

Omri Tau[1,5], Alice Henley [1,5], Anton N. Boichenko[2], Nadezhda N. Kleshchina[2], River Riley[1], Bingxing Wang [1,4], Danielle Winning[1], Ross Lewin[1], Ivan P. Parkin[1], John M. Ward [3], Helen C. Hailes [1], Anastasia V. Bochenkova [2✉] & Helen H. Fielding [1✉]

Green fluorescent protein (GFP), the most widely used fluorescent protein for in vivo monitoring of biological processes, is known to undergo photooxidation reactions. However, the most fundamental property underpinning photooxidation, the electron detachment energy, has only been measured for the deprotonated GFP chromophore in the gas phase. Here, we use multiphoton ultraviolet photoelectron spectroscopy in a liquid-microjet and high-level quantum chemistry calculations to determine the electron detachment energy of the GFP chromophore in aqueous solution. The aqueous environment is found to raise the detachment energy by around 4 eV compared to the gas phase, similar to calculations of the chromophore in its native protein environment. In most cases, electron detachment is found to occur resonantly through electronically excited states of the chromophore, highlighting their importance in photo-induced electron transfer processes in the condensed phase. Our results suggest that the photooxidation properties of the GFP chromophore in an aqueous environment will be similar to those in the protein.

[1] Department of Chemistry, University College London, 20 Gordon Street, London WC1H 0AJ, UK. [2] Department of Chemistry, Lomonosov Moscow State University, 119991 Moscow, Russia. [3] The Advanced Centre for Biochemical Engineering, Department of Biochemical Engineering, University College London, Gower Street, London WC1E 6BT, UK. [4] Present address: College of Chemistry and Chemical Engineering, Henan Institute of Science and Technology, Eastern Hualan Avenue, Xinxiang 453003, China. [5] These authors contributed equally: Omri Tau and Alice Henley. ✉email: bochenkova@phys.chem.msu.ru; h.h.fielding@ucl.ac.uk

Many processes in nature rely on photoactive proteins to transform light energy into a physical response, for example, photosynthesis and vision. At the heart of these processes is a small chromophore that absorbs light and undergoes small-scale structural changes, which are captured and amplified by the surrounding protein and used to initiate a macromolecular-level response and, in turn, biological function. Understanding the photophysics and photochemistry behind the selectivity and efficiency of nature's photoactive proteins is crucial for the rational design of new photomaterials for a range of applications including photovoltaics and bioimaging. Although a detailed knowledge of the intrinsic electronic structure of photoactive protein chromophores is a good starting point, it is essential to understand the role an environment plays in tuning the electronic structure. Experimentally, one of the most direct ways of probing electronic structure is through the measurement of electron binding energies using photoelectron spectroscopy (PES). However, although there have been numerous PES studies of protein chromophores in the gas phase[1], PES studies of protein chromophores in more complex environments have not yet been reported.

Green fluorescent protein (GFP) has revolutionised the life sciences by enabling a wide range of applications such as fluorescence imaging and biosensing[2–9]. The chromophore that lies at the heart of GFP, 4-hydroxybenzylidene-1,2-dimethylimidazolinone (p-HBDI), is anchored covalently and by a network of hydrogen bonds to the protein that is wrapped around it in a β-barrel structure. GFP has two absorption bands, centred around 395 nm and 480 nm, that are associated with the neutral and deprotonated anionic forms of the chromophore, respectively. Its characteristic green emission (~509 nm) comes from the anionic form of the chromophore (p-HBDI⁻, Fig. 1 (inset)) with high quantum yield, $\Phi = 0.79$[10].

The first absorption bands in the electronic absorption spectra of GFP and the isolated p-HBDI⁻ chromophore in vacuo lie remarkably close to one another[11–15], suggesting that the β-barrel structure provides an electronic environment that is similar to a vacuum; however, the environment of the chromophore plays a crucial role in defining the electronic relaxation dynamics

following photoexcitation of the first electronically excited state. Denaturation of the protein results in a loss of fluorescence[16] and the isolated chromophore is virtually non-fluorescent in vacuo[12] and at physiological temperatures in solution[17], although it has a sufficiently long excited-state lifetime in vacuo when cooled to 100 K that it is expected to fluoresce[18], and it fluoresces strongly in solution when cooled to 77 K[17]. The lack of fluorescence from the free chromophore at physiological temperatures has been attributed to ultrafast double-bond isomerisation, followed by internal conversion back to the electronic ground state[19,20], with timescales that are similar in the gas phase[21] and in solution[19,20], suggesting that the intrinsic electronic properties of the chromophore determine the relaxation mechanism.

Higher lying electronically excited states of GFP are believed to be involved in photooxidation processes[22] and in the formation of solvated electrons[23]. However, the electron detachment energy, the most fundamental property underpinning photooxidation, and the electron binding energies of the higher lying electronic states have only been determined experimentally for the deprotonated GFP chromophore in the gas phase[11,13,21,24–28]. Here, we report the results of multiphoton (MP) ultraviolet (UV) liquid-microjet PES experiments and high-level quantum chemistry calculations of p-HBDI⁻ in aqueous solution. Measurements of the electron kinetic energy (eKE) distribution following photodetachment at a range of wavelengths allow us to determine the first three vertical detachment energies (VDEs) to an accuracy of ±0.2 eV, representing uncertainties in the effect of inelastic scattering of low energy electrons (< 5 eV) and the vacuum level offset (Supplementary Methods). Our results suggest that the photooxidation properties of the deprotonated GFP chromophore in aqueous solution are similar to those of the chromophore in its natural protein environment.

## Results

**UV–vis absorption spectrum.** The UV–vis absorption spectrum of aqueous p-HBDI⁻ is presented in Fig. 1 together with the vertical excitation energies (VEEs) calculated using Extended Multi-configuration Quasi-degenerate Perturbation Theory (XMCQDPT2)[29,30] and Effective Fragment Potential (EFP)[31] methods.

The absorption maximum of aqueous p-HBDI⁻ is 422 nm (2.94 eV), whereas that of gas-phase p-HBDI⁻ is 482 nm (2.57 eV)[11], similar to the absorption maximum of the protein in its anionic form (480 nm, 2.58 eV). Below 350 nm, there are a number of small peaks in the UV–vis absorption spectrum of aqueous p-HBDI⁻, corresponding to transitions to higher-lying electronically excited states. The VEEs of these higher-lying states are all blue-shifted compared to those of the higher-lying electronically excited states of gas-phase p-HBDI⁻[13,28].

**Photoelectron spectra.** Figure 2a shows the one-colour MP photodetachment spectra of aqueous p-HBDI⁻ using 440 nm, 422 nm and 403 nm (resonant with the $S_0$-$S_1$ transition, with increasing excess energy in $S_1$), 300 nm (close to the absorption minimum) and 249.7 nm (resonant with higher-lying electronically excited states). The photoelectron spectra are plotted as a function of electron kinetic energy (eKE), corrected to account for the spectrometer transmission function, and fitted with sums of Gaussians. To account for uncertainty in the transmission function for very low eKE, we only fit to measured eKE > 0.3 eV (Supplementary Methods) which, after correcting for the vacuum level offset between the interaction region and the analyser (Supplementary Methods), corresponds to true eKE > 0.4 eV. The centres of the fitted Gaussians are approximated to vertical detachment processes and are listed in Table 1 together with the

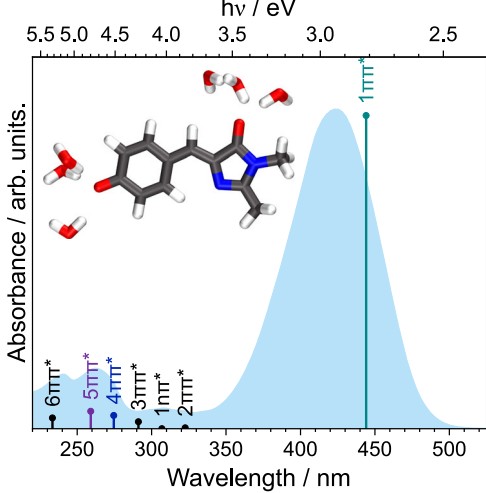

**Fig. 1 UV–vis absorption spectrum of 20 μM aqueous solution of p-HBDI⁻ (blue).** Vertical lines mark XMCQDPT2/SA(10)-CASSCF(16,14)/(aug)-cc-pVDZ//EFP calculated vertical excitation energies (VEEs) with heights proportional to oscillator strengths. Inset: PBE0/(aug)-cc-pVDZ//EFP(253) equilibrium geometry of p-HBDI⁻ + 6H₂O. In the depiction of the equilibrium geometry, carbon atoms are shown in black, oxygen atoms in red, nitrogen atoms in blue and hydrogen atoms in white.

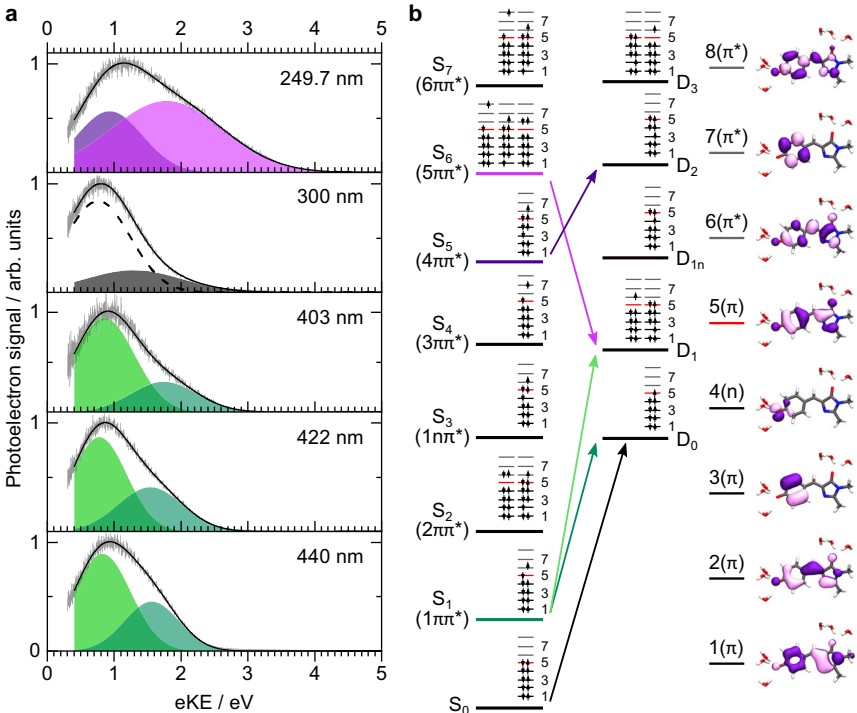

**Fig. 2 Photoelectron spectra of aqueous *p*-HBDI⁻ and electronic configurations and molecular orbitals of aqueous *p*-HBDI⁻. a** MP detachment photoelectron spectra (grey lines) of 20 *µ*M aqueous solution of *p*-HBDI⁻ recorded following photoexcitation at 440 nm, 422 nm, 403 nm, 300 nm, and 249.7 nm, plotted as a function of eKE. Fitted Gaussians represent $S_1$-$D_0$ (dark green), $S_1$-$D_1$ (light green), $S_0$-$D_0$ (grey, see text), $S_4 \to e^-_{(aq)} \to e^-_{(g)}$ (dashed line, see text), $S_6$-$D_1$ (pink), and $S_5$-$D_2$ (purple) detachment processes. **b** Left: electronic configurations and molecular orbitals of the first few electronic states of *p*-HBDI⁻ and the corresponding neutral radical in aqueous solution. Arrows highlight the detachment processes contributing to the photoelectron spectra presented. Right: natural *π*- and n-type orbitals obtained using CASSCF(16,14) with the pure *π* and mixed n/*π* active spaces, respectively. In the depiction of molecular geometries, carbon atoms are shown in black, oxygen atoms in red, nitrogen atoms in blue and hydrogen atoms in white.

**Table 1 Experimental vertical detachment energies (VDEs) of *p*-HBDI⁻ in aqueous solution. Measured centres of Gaussians fitted to the photoelectron spectra in Fig. 2a, eKE, associated values of VDE = $nh\nu$ − eKE, where *n* is the number of photons required to detach the electron from the assigned singlet state, assignments and corresponding $S_0$-$D_i$ VDEs (see Discussion). The error bars associated with the measured eKEs reflect experimental uncertainties (Supplementary Methods).**

| Wavelength/ nm (eV) | eKE/eV | n | VDE/eV | Assignment | $S_0$-$D_i$ VDE/eV |
|---|---|---|---|---|---|
| 440 (2.82) | 0.8 ± 0.2 | 2 | 4.8 ± 0.2 | $S_1$-$D_1$ | 7.6 ± 0.2 |
|  | 1.6 ± 0.2 | 2 | 4.0 ± 0.2 | $S_1$-$D_0$ | 6.8 ± 0.2 |
| 422 (2.94) | 0.8 ± 0.2 | 2 | 5.1 ± 0.2 | $S_1$-$D_1$ | 7.9 ± 0.2 |
|  | 1.5 ± 0.2 | 2 | 4.4 ± 0.2 | $S_1$-$D_0$ | 7.2 ± 0.2 |
| 403 (3.08) | 0.9 ± 0.2 | 2 | 5.3 ± 0.2 | $S_1$-$D_1$ | 8.1 ± 0.2 |
|  | 1.7 ± 0.2 | 2 | 4.5 ± 0.2 | $S_1$-$D_0$ | 7.3 ± 0.2 |
| 300 (4.13) | 0.8 ± 0.2 | 1 | 3.3 ± 0.2 | — | — |
|  | 1.3 ± 0.2 | 2 | 7.0 ± 0.2 | $S_0$-$D_0$ | 7.0 ± 0.2 |
| 249.7 (4.97) | 0.9 ± 0.2 | 1 | 4.1 ± 0.2 | $S_5$-$D_2$ | 8.6 ± 0.2 |
|  | 1.8 ± 0.2 | 1 | 3.2 ± 0.2 | $S_6$-$D_1$ | 8.0 ± 0.2 |

corresponding vertical detachment energies, VDE = $nh\nu$ − eKE, where *n* is the number of photons in the detachment process. The fact that the measured photoelectron spectra can be fit so well with Gaussians suggests that the spectral profiles have not been distorted significantly by inelastic scattering from solvent molecules as they leave the liquid. This is consistent with

photoelectrons being emitted from weakly-soluble organic molecules with enhanced surface concentration[32]. Our simulations of electron trajectories suggest that photoelectrons generated within a few nanometres of the surface of a liquid-microjet lose minimal energy to inelastic scattering and that any loss is within experimental error (Supplementary Methods).

**Electronic structure**. The MP detachment processes giving rise to each photoelectron spectrum can be deduced by considering the energetics of all the processes contributing to the five photoelectron spectra. This requires an understanding of the electronic structure of *p*-HBDI⁻ and its corresponding neutral radical. The VEEs of the first seven excited singlet states of *p*-HBDI⁻ in aqueous solution have been determined using high-level XMCQDPT2/SA(10)-CASSCF(16,14)/(aug)-cc-pVDZ//EFP calculations and are listed in Table 2. The first VDE has been calculated using a hybrid DFT/EFP/MD approach and depends strongly on the system size as it relies on the accuracy of the absolute energy estimation of long-range interactions between the negatively charged chromophore and polar water molecules. It converges for a system with ~12,500 water molecules (*R* = 40 Å, Supplementary Fig. 8). The equilibrium geometry of this system is then used to calculate the lowest-lying $D_0$ and $D_1$ VDEs using the XMCQDPT2/SA(10)-CASSCF(14,14)/(aug)-cc-pVDZ+//EFP approach. The calculated VEEs are affected less by the size as the system is still negatively charged after photoexcitation. VDEs of the higher-lying doublet states have been calculated at the XMCQDPT2/SA(10)-CASSCF(15,14)/(aug)-cc-pVDZ//EFP level of theory with respect to $D_0$. The VDEs are also listed in Table 2.

Table 2 XMCQDPT2/EFP calculated vertical excitation energies (VEEs) and oscillator strengths (*f*) of *p*-HBDI$^-$ in aqueous solution. The lowest-lying doublet states, $D_0$ and $D_1$, are calculated for a system embedded in a water sphere with $R = 40$ Å, using the hybrid DFT/EFP/MD equilibrium geometry shown in the Supplementary Methods, at the XMCQDPT2/EFP level of theory. The higher-lying vertical detachment energies (VDEs) are estimated with respect to $D_0$ at the minimum energy geometry of the anion.

| Excited state | $S_1/1\pi\pi^*$ | $S_2/2\pi\pi^*$ | $S_3/1n\pi^*$ | $S_4/3\pi\pi^*$ | $S_5/4\pi\pi^*$ | $S_6/5\pi\pi^*$ | $S_7/6\pi\pi^*$ |
|---|---|---|---|---|---|---|---|
| VEE/eV | 2.79 | 3.84 | 4.04 | 4.26 | 4.51 | 4.79 | 5.32 |
| *f* | 1.133 | 0.003 | 0.000 | 0.025 | 0.048 | 0.063 | 0.039 |
| **Detached state** | $D_0$ | $D_1$ | $D_{1n}$ | $D_2$ | $D_3$ | $D_4$ | $D_5$ |
| VDE/eV | 6.7 | 7.7 | 8.6 | 8.8 | 9.4 | 9.8 | 10.8 |

To a first approximation, we can use Koopmans' theorem to determine detachment propensities. For example, from the electronic configurations shown in Fig. 2b, we see that the $S_0$ state is most likely to ionise to $D_0$, $S_1$ to $D_0$ and $D_1$, $S_5$ to $D_2$ and $S_6$ to $D_1$. These propensities, together with conservation of energy considerations, allow us to determine the MP detachment processes contributing to the photoelectron spectra presented in Fig. 2a.

## Discussion

Since the UV absorption cross-section is very low at 300 nm (Fig. 1), we expect direct detachment from the ground electronic state to have a significant contribution to the photoelectron spectrum (Fig. 2a). The 300 nm spectrum is best fit with two Gaussians with maximum eKEs of $1.3 \pm 0.2$ eV (dark grey) and $0.8 \pm 0.2$ eV (dashed line). The corresponding two-photon VDEs from $S_0$ are $7.0 \pm 0.2$ eV and $7.5 \pm 0.2$ eV, respectively. From Koopmans' considerations, we assign the feature corresponding to $7.0 \pm 0.2$ eV as the $S_0$-$D_0$ VDE. This is around 4 eV higher than that of gas-phase *p*-HBDI$^-$ (2.73 eV)[33] and over 3 eV higher than *p*-HBDI$^-$ microsolvated by one or two water molecules (3.15 and 3.50 eV, respectively)[34]. The measured VDE is in good agreement with our calculated value (6.7 eV) and similar to an earlier EOM-CCSD/EFP calculation (6.6 eV)[35]. It is worth noting that the $S_4$ state located around 4.26 eV (Table 2) could be involved in resonance-enhanced $1 + 1$ electron detachment to $D_0$ at this photon energy (4.13 eV). Koopmans' arguments indicate that the feature corresponding to $7.5 \pm 0.2$ eV is unlikely to arise from resonant $1 + 1$ detachment to a higher lying VDE. However, two-photon direct $S_0$-$D_1$ detachment may contribute to this peak, as a second-order non-linear process (Table 2). At the same time, it is possible that the $S_4$ state is involved in resonant $1 + 1$ electron detachment via formation of a solvated electron, which is discussed in more detail below. Although it is possible to fit the 300 nm spectrum with a single Gaussian with eKE ~ 0.6 eV Supplementary Methods, the improved fit with two Gaussians suggests that the processes giving rise to the spectrum are best described by fitting two Gaussians.

The 440 nm, 422 nm and 403 nm photoelectron spectra were recorded following photoexcitation of the $S_1(1\pi\pi^*)$ state. All three photoelectron spectra have similar profiles and are best fit with two Gaussians. Knowing the $S_0$-$D_0$ VDE allows us to deduce that these spectra arise from $1 + 2$ resonantly enhanced processes. For the 440 nm photoelectron spectrum, the Gaussian with eKE = $1.6 \pm 0.2$ eV (dark green) corresponds to a two-photon VDE of $4.0 \pm 0.2$ eV and the Gaussian with eKE = $0.8 \pm 0.2$ eV (light green) corresponds to a two-photon VDE of $4.8 \pm 0.2$ eV. To assign these features, we assume that solvent reorganisation is negligible on the timescale of ionisation, that vibrational energy is conserved during detachment, and estimate the $S_0$-$D_i$ VDE

(Table 1) using,

$$\text{VDE}(S_0 - D_i) \approx mh\nu - (h\nu - \text{AEE}) - \text{eKE}, \quad (1)$$

where $h\nu$ is the photon energy, $m$ is the total number of photons involved in the MP detachment process ($m = 3$ for the 440 nm, 422 nm and 403 nm photoelectron spectra) and AEE is the adiabatic excitation energy (in this case, the $S_0$-$S_1$ AEE). The calculated reorganisation energy is only 0.1 eV in $S_1$ so we assume that the AEEs are similar to VEEs (Table 2) within experimental error. For the feature with eKE = $1.6 \pm 0.2$ eV (dark green), the $S_0$-$D_n$ VDE is $6.8 \pm 0.2$ eV, which is in good agreement with our measured value for the $S_0$-$D_0$ VDE obtained from the 300 nm photoelectron spectrum ($7.0 \pm 0.2$ eV) and in excellent agreement with our calculated value. For the feature with eKE = $0.8 \pm 0.2$ eV (light green), the $S_0$-$D_i$ VDE is $7.6 \pm 0.2$ eV; based on Koopmans' considerations, this can be assigned as the $S_0$-$D_1$ VDE. Our calculated $S_0$-$D_1$ (7.7 eV) VDE is consistent with this assignment. For the 422 nm spectrum, we use similar arguments to obtain $S_0$-$D_0$ and $S_0$-$D_1$ VDEs of $7.2 \pm 0.2$ eV and $7.9 \pm 0.2$ eV. For the 403 nm spectrum, we use similar arguments to obtain $S_0$-$D_0$ and $S_0$-$D_1$ VDEs of $7.3 \pm 0.2$ eV and $8.1 \pm 0.2$ eV. It is clear that the measured $S_0$-$D_0$ and $S_0$-$D_1$ VDEs increase with increasing excess vibrational energy in $S_1$ (Table 1 and Supplementary Note 1). We attribute this to relaxation on $S_1$ occurring on the timescale of the measurement (~ 150 fs). This is consistent with the 210 fs timescale observed for ultrafast double-bond isomerisation of *p*-HBDI$^-$ in aqueous solution, determined from ultrafast fluorescence measurements[19]. Relaxation on $S_1$ would be expected to result in an effective increase in the $S_1$-$D_0/D_1$ VDEs[21], which would lead to the $S_0$-$D_0/D_1$ VDEs determined using Eq. (1) being overestimated. As such, the VDEs determined from the 440 nm photoelectron spectrum are presumed to be more accurate than those recorded at 422 nm and 403 nm, and we note that both the $S_0$-$D_0$ and $S_0$-$D_1$ VDEs determined from the 440 nm spectrum are in excellent agreement with those obtained from theory (Supplementary Note 1).

The 249.7 nm photoelectron spectrum was recorded following photoexcitation of higher-lying electronically excited states and is best fit with two Gaussians. The maxima of the Gaussians have eKEs at $1.8 \pm 0.2$ eV (pink) and $0.9 \pm 0.2$ eV (purple), corresponding to one-photon VDEs of $3.2 \pm 0.2$ eV and $4.1 \pm 0.2$ eV, respectively. Using Eq. (1), with $m = 2$ (total number of photons to ionise from the ground state), for the feature with eKE = $1.8 \pm 0.2$ eV, we estimate $S_0$-$D_i$ VDEs of $8.0 \pm 0.2$ for $1 + 1$ detachment via the $S_6$ state, which is located at 4.8 eV and has an appreciable oscillator strength (Table 2). We assign this feature to $S_6$-$D_1$ detachment. The corresponding $S_0$-$D_1$ VDE is consistent with those obtained from the 440 nm spectrum (Table 1, Supplementary Note 1). For the feature with eKE = $0.9 \pm 0.2$ eV, we estimate an $S_0$-$D_i$ VDE of $8.6 \pm 0.2$ eV for $1 + 1$ detachment via the $S_5$ state, which correlates with the $D_2$ state. We assign this

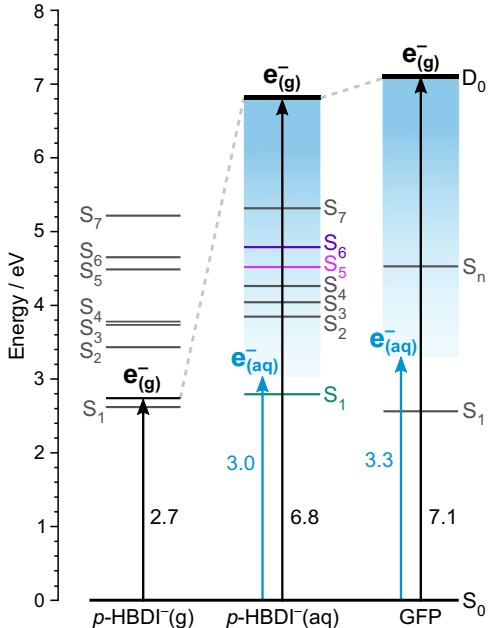

**Fig. 3 Schematic energy level diagram showing the electronic structure of $p$-HBDI$^-$ in the gas phase (left), in aqueous solution (centre) and in S65T-GFP (right).** Black arrows represent vertical detachment energies (VDEs). Blue arrows indicate the thresholds for formation of solvated electrons in water, taking VDE = 3.76 eV[36]. Vertical excitation energies (VEEs) and VDEs for aqueous $p$-HBDI$^-$ are from this work and those for gas-phase $p$-HBDI$^-$ and the S65T-GFP protein are taken from Ref. [28]. The VEE of the higher lying electronically excited gateway state proposed to be involved in the formation of solvated electrons in the protein is labelled $S_n$.

feature to $S_5$-$D_2$ detachment. Our calculated $S_0$-$D_2$ (8.8 eV) VDE is consistent with this assignment.

It has been proposed that one of the higher-lying $\pi\pi^*$ states is a gateway state for resonant electron transfer to solvent[28]. Specifically, the $S_4$ state has the potential to undergo autodetachment to $D_0$, since it is a one-electron process. The VEEs and lowest VDE of aqueous $p$-HBDI$^-$ are plotted in Fig. 3, together with those of the chromophore in the gas phase and in the S65T-GFP protein[28]. Although the VEEs of the first electronically excited singlet state are very similar in the gas phase and in the protein, the VDEs are very similar in aqueous solution and in the S65T-GFP protein. The VDE of a fully solvated electron, $e^-_{(aq)} \rightarrow e^-_{(g)}$, has been determined to be 3.76 eV in water[36]. Therefore, the higher-lying electronically excited states of $p$-HBDI$^-$ are embedded in the continuum of the solvated electron in water and in the S65T-GFP protein. This suggests that the photooxidation properties of the anionic form of S65T-GFP and those of aqueous $p$-HBDI$^-$ will be similar.

Solvated electrons could be formed following $S_0$-$S_4$ photo-excitation (VEE 4.26 eV) and autodetachment out of the $S_4$ state. In principle, autodetachment from the $S_4$ state will be fast because it is an excited-shape resonance with respect to the quasi-continuum of the solvated electron. If it is faster than the time-scale of our measurement, which is defined by the duration of our laser pulses (~150 fs), and if the processes contributing to the 300 nm (4.13 eV) photoelectron spectrum are best interpreted by fitting two Gaussians, the feature with eKE = $0.8 \pm 0.2$ eV could be assigned to the $e^-_{(aq)} \rightarrow e^-_{(g)}$ process. $VDE_{e^-} \approx h\nu - eKE = 3.3 \pm 0.2$ eV, which is consistent with that of a partially solvated electron[37]. Transient electronic absorption spectroscopy measurements of the phenolate moiety of $p$-HBDI$^-$ in aqueous solution propose electron appearance timescales of a few hundred

femtoseconds[38] to picoseconds[39]. We plan time-resolved photo-electron spectroscopy measurements to confirm if solvated electrons are formed following photoexcitation of the higher-lying state of solvated $p$-HBDI$^-$.

In summary, we have employed MP UV photoelectron spectroscopy in a liquid-microjet to measure the VDE of the deprotonated GFP chromophore in aqueous solution. To the best of our knowledge, this is the first reported VDE measurement of any protein chromophore and highlights the value of MP UV photoelectron spectroscopy for probing the electronic structure of sparingly soluble biological chromophores. Notably, the first VDE ($6.8 \pm 0.2$ eV) is more than double that of the deprotonated chromophore *in vacuo* ($2.73 \pm 0.01$ eV)[33] as a result of solvent-stabilisation of the anion. Although the VEE of the first electronically excited singlet state of the protein in its anionic form is very similar to that of the deprotonated chromophore *in vacuo*, the VDE and the pattern of higher lying electronically excited states in the protein are similar to those of the deprotonated chromophore in aqueous solution. We propose that the photo-oxidation properties of the deprotonated chromophore in aqueous solution are similar to those of the protein and propose that higher-lying excited states of solvated $p$-HBDI$^-$ act as a gateway for electron transfer processes in the condensed phase.

## Methods

**Experimental**. $p$-HBDI was prepared using reported procedures[40].

Photoelectron spectra were recorded using our liquid-microjet magnetic-bottle photoelectron spectrometer[41]. 20μM of $p$-HBDI was dissolved in water. 5 mM NaOH was added to deprotonate $p$-HBDI to form $p$-HBDI$^-$ and to reduce charging of the microjet nozzle. UV–vis spectra of the solution were recorded before and after each experiment to ensure that $p$-HBDI was in its deprotonated anionic form. The $p$-HBDI$^-$ solution was introduced into the photoelectron spectrometer through a 20 μm diameter fused silica capillary using a high-performance liquid chromatography pump operating with a backing pressure of ~85 bar and a flow-rate of 0.7 ml/min. The $p$-HBDI$^-$ solution was intersected with femtosecond laser pulses 2 mm downstream from the capillary nozzle before being collected and recirculated. Photoelectrons were detected at the end of a time-of-flight (TOF) tube and the photoelectron current was recorded together with the arrival time relative to the trigger of the laser pulse. Photoelectron spectra of NO and Xe were recorded to convert TOF to eKE and the energy resolution ($\Delta E/E \sim 1\%$) and to determine the instrument function, streaming potential and vacuum level offset between the interaction region and analyser, the procedures for which are described in detail in the Supplementary Methods.

**Computational**. Ground-state geometries of the solvated chromophore were obtained using a hybrid QM/EFP/MD approach described in the Supporting Information. The quantum mechanical part included a single $p$-HBDI$^-$ anion with six water molecules closest to the chromophore and was described at the PBE0/(aug)-cc-pVDZ level of theory. The first VDE was calculated as the energy difference between the $S_0$ and $D_0$ states in the geometry of the anion at the PBE0/(aug)-cc-pVDZ//EFP level of theory for a series of the model systems of increasingly larger size with up to 19,000 water molecules (Supplementary Fig. 7). All water molecules, except for those included in the quantum mechanical part, were treated as EFPs. A structure with ~12,500 water molecules ($R = 40$ Å) was then used to calculate the $D_0$ and $D_1$ VDEs using the XMCQDPT2/ SA(10)-CASSCF (14,14)/(aug)-cc-pVDZ+//EFP method, with a very diffuse function placed outside the water sphere and included in the active space to mimic electron detachment.

A smaller-sized pure QM/EFP system with 253 water molecules was used for excited-state calculations (Supplementary Fig. 9). Vertical excitation energies in the solvated anion and neutral radical were calculated at the XMCQDPT2/ CASSCF(16,14)//EFP and XMCQDPT2/CASSCF(15,14)//EFP levels of theory, respectively, using the same basis set. The $S_0$–$S_1$ absorption profile for a single configuration of the solvated $p$-HBDI$^-$ anion and the difference between the vertical and adiabatic excitation energies, i.e., the reorganisation energy in $S_1$, were calculated under the double harmonic parallel-mode approximation, using the XMCQDPT2/CASSCF(14,13)//EFP excited-state gradient calculated at the Franck-Condon point. Vibrational analysis was performed in the ground state using the PBE0/(aug)-cc-pVDZ//EFP method. For all electronic structure calculations, the Firefly computational package[30] was used.

## Data availability

The raw data and energy-calibrated data obtained from experiments and Cartesian coordinates for all structures employed in the computational calculations have been deposited in the UCL Research Data Repository at https://doi.org/10.5522/04/17295371.

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

## Acknowledgements

This work was supported by the EPSRC (EP/L005646/1), the Diamond Light Source (STU0157) and the Royal Society and Leverhulme Trust (SRF/R1/180079). A.N.B., N.N.K. and A.V.B. acknowledge support from the Russian Science Foundation (Project No. 17-13-01276). Calculations were carried out using HPC services provided by the Department of Chemistry at UCL managed by Dr Frank Otto, the shared research facilities of HPC computing resources at Lomonosov Moscow State University and local resources provided through the Lomonosov Moscow State University Program of Development.

## Author contributions

H.H.F. conceived the project. The experimental data reported here were recorded by O.T. and analysed by O.T. with assistance from A.H., to improve upon data first recorded and analysed by R.R. with assistance from A.H., B.W. and D.W. Scattering similations were performed by A.H. and quantum chemistry calculations were performed by A.N.B., N.N.K., A.H. and A.V.B. *p*-HBDI was synthesised by R.L under the supervision of J.M.W. and H.C.H. The data reported here were analysed by A.H. and O.T. Results were interpreted by A.H., R.R., A.V.B. and H.H.F. The manuscript was written by H.H.F. with contributions from A.H., A.V.B., R.R. and O.T. and comments from I.P.P.

## Competing interests

The authors declare no competing interests.
