## [Peer Review File · Nature Communications]

REVIEWER COMMENTS

Reviewer #1 (Remarks to the Author):

Riley et al. report a combined experimental and computational study on the photoelectron spectrum of the pHBDI anion (the GFP chromophore). Experiments are done at several wavelengths and nicely supplemented by state-of-the-art computational methods that are used to assign the bands in the spectra. The computational models employ a combination of multi-reference methods and an EFP solvation model. A key quantity, the S0-D0 vertical detachment energy, is computed with density functional theory but this is reasonable/necessary given that detachment/ionization energy calculations are sensitive to the solvent box size and require a very large solvent box (this computational-cost related choice needs to be better highlighted in the manuscript).

The manuscript is recommended for publication basically as it is. I have just a few comments and questions that may generate a manuscript minor revision:

- I am not sure I follow the argument about the solvated electron. Is the hypothesis that the light grey peak in the 300 nm spectrum the result of S0-S4 excitation -> auto-detachment to form a solvated electron -> photoionization of the solvated electron? Surely, such a process would necessarily be sequential, since the S4 auto-detaching state would likely have a finite lifetime, and the entire process could therefore not be directly probed with MP spectroscopy? I can only imagine such a state can be probed, as the authors suggest towards the end of the manuscript, with a time resolved (i.e., sequential) pump-probe experiment. Maybe, as per Occam's razor, a simpler explanation exists. For instance, have the authors tried to fit the 300 nm photoelectron spectrum using only one Gaussian? Or, other states that were left out from the analysis on the basis of Koopmans' theorem may have some small contributions as well?

- To be clear, when the authors mention using an "energy difference method" with PBE0, does that mean calculating the S0 and D0 for the same geometry + environment at the PBE0 level of theory and then finding the energy difference?

- The approach used to determine the effect of inelastic scattering (Δ_{meas}) is unclear. It would be good to add more details in the SI. I was particularly confused by Fig. S2. Why is the "used" correction in this study sometimes taken from the water plot, and sometimes in between the water and methanol plots?

Minor points:

Page 2 top. I think the following change: "macroscopic response" -> "macromolecular-level response and, in turn, biological function" is necessary.

Page 2 bottom. "the most direct way" -> "one of the most direct way"

Page 3 middle. "rapid isomerization" - "ultrafast double-bond isomerization".

Reviewer #2 (Remarks to the Author):

This manuscript describes measurements of photoelectron spectra of the green fluorescent protein (GFP) chromophore in a water-methanol solution using a liquid microjet. The spectra are measured by multiphoton ultraviolet photoelectron spectroscopy. Both resonant and non-resonant pathways are identified and the vertical detachment energies (VDEs) corresponding to the ground and several electronically excited states of both the anion and the neutral molecule are reported. The experimental results are compared with QM/MM calculations relying on density-functional theory, the effective-fragment potential method and a very large number of water molecules to reach converged detachment energies.

This study is very interesting and important because it demonstrates the capability of liquid microjet photoelectron spectroscopy to access detachment (or ionization) energies of protein chromophores, even when they have rather low solubilities. The GFP chromophore is of particular interest for many biochemical applications and is therefore an excellent choice. Overall, the study has been carried out with care, the experimental data recorded at six different wavelengths of the detachment laser are very detailed and the performed non-linear fits are convincing. The theory effort is also particularly impressive, especially given the size and quality of the performed calculations, the demonstrated convergence of the energies with increasing numbers of water molecules and the agreement with the experimental results.

However, the main result of this work are absolute VDEs and their comparison with theory. The determination of detachment and ionization energies from liquid-microjet photoelectron spectroscopy is an important yet still not fully understood topic that has been the object of many

recent publications regarding, in particular, the correct referencing of absolute energies and the effect of inelastic electron scattering.

Specifically, I would like the authors to address the following comments:

1) Inelastic scattering of electrons in water/methanol has been convincingly shown to modify the observed photoelectron spectra in Refs. 33 and 34. The authors have made use of these results to extract a correction “D_meas” (see Table 1 and Fig. S2). Inelastic scattering can cause energy loss, i.e. a positive D_meas, but it is unclear how the authors justify the negative values of D_meas up to measured kinetic energies of 1 eV. This aspect is not discussed in the present work and it appears to be unphysical.

2) Recent work on photoelectron spectroscopy (PES) of static liquids has shown that, similar to solid-state PES, the Fermi levels of a conductive solution and the photoelectron analyzer are equilibrated (Tissot et al., *Top. Catal.* 59, 605, 2016). Since the work functions of the solution and the analyzer are different, this situation results in an energetic offset of the vacuum levels, which will lead to incorrect absolute energies unless it is compensated. In solid-state PES, this effect is well known and addressed by biasing the sample. Similar ideas have been applied to liquid microjets by Olivieri et al., *PCCP* 18, 29506, 2016 and by Perry et al., *JPC Lett.* 11, 1789, 2020, who found significant deviations from ionization energies reported earlier. In the present work, the authors briefly mention the streaming potential, but they do not address the issue of energy referencing, which is important when reporting absolute energies.

3) The authors used 30 mM NaF to compensate for the effects of the streaming potential. They do not justify this choice, nor do they show that the streaming potential is effectively compensated. This would, however, be an important and straightforward experimental verification that would contribute to support the reported VDEs.

4) The transmission function of photoelectron spectrometers, including magnetic-bottle devices, tends to decrease significantly at very low kinetic energies. This, in turn, will affect the determination of the VDEs involving the slowest electrons because the corresponding line shapes will not be Gaussian. Have the authors verified the assumed energy independence of the low-energy electron transmission function (e.g. by gas-phase measurements on NO)?

Reviewer #3 (Remarks to the Author):

This paper presents photoelectron spectrum of the GFP chromophore in a liquid water microjet. The main objective is to determine the vertical detachment energy (VD) of the chromophore, and this is found to be 6.8 eV, about 4 eV higher than the VDE of the gas phase species. The higher VDE is consistent with the results for many aqueous anions. While the determination of this value is of some interest, the main result of the paper seems pretty straightforward and there is little reason to publish in a high profile journal such as Nature Comm.

The authors have done a thorough job in figuring out wavelength dependence of their photoelectron spectra; this complex dependence results from the energetic requirement of two photons to detach an electron of the solvated chromophore. One thus has to sort out the non-resonant vs resonant multiphoton processes. It would have been much easier to do this experiment in a synchrotron where one-photon detachment would be energetically accessible.

December 16, 2020

Reviewer 1

Riley et al. report a combined experimental and computational study on the photoelectron spectrum of the pHBDI anion (the GFP chromophore). Experiments are done at several wavelengths and nicely supplemented by state-of-the-art computational methods that are used to assign the bands in the spectra. The computational models employ a combination of multi-reference methods and an EFP solvation model. A key quantity, the S_0 - D_0 vertical detachment energy, is computed with density functional theory but this is reasonable/necessary given that detachment/ionization energy calculations are sensitive to the solvent box size and require a very large solvent box (this computational-cost related choice needs to be better highlighted in the manuscript).

In addition to the DFT calculations used to estimate the S_0 - D_0 VDE and to test the convergence of the calculated VDE with respect to the system size, we have now calculated the S_0 - D_0 VDE using multistate multireference perturbation theory XMCQDPT2 coupled to the EFP method. We have previously shown that the calculated VDE is strongly dependent on the system size, reaching the converged value of 6.84 eV at the PBE0/(aug)-cc-pVDZ/EFP level of theory for a system with $\sim 12,500$ water molecules ($R = 40 \text{ \AA}$). The equilibrium geometry of this hybrid PBE0/EFP/MD system with ~ 1500 water molecules included in the PBE0/ EFP core is now used for a single-point energy calculation of the S_0 - D_0 VDE using the XMCQDPT2/EFP(12,550) method.

The VDE is calculated using the (aug)-cc-pVDZ basis set, which is augmented by very diffuse functions of the p-type with a 10^{-10} exponent (IP orbitals). The IP functions are placed at a distance of 40 \AA from the edge of the water sphere along the direction of the dipole moment of the *p*-HBDI radical. The calculated VDE is found to be insensitive to the position of the IP orbitals outside the water sphere for distances larger than 30 \AA . One of these IP orbitals is included in the active space to mimic electron detachment. In addition to the IP orbital, all valence π orbitals, except for the lone pair orbital localized on the nitrogen atom of the imidazolinone ring, are included in the active space, which results in 14 electrons distributed over 14 orbitals. The XMCQDPT2 effective Hamiltonian is constructed in the frame of the model space spanned by 10 state-averaged CASSCF(14,14) wavefunctions. The ground-state energies of the *p*-HBDI anion (S_0) and its radical (D_0) with an ejected electron occupying the IP orbital are then obtained in a single XMCQDPT2/EFP calculation. The quantum mechanical part includes *p*-HBDI⁻ and six water molecules closest to the chromophore, while all other water molecules are treated using the EFP method.

The large-scale XMCQDPT2/EFP calculations give the S_0 - D_0 VDE of 6.75 eV, which is consistent with the value estimated using the PBE0/EFP method (6.84 eV). The VDE of 6.8 eV, which has been used for interpreting the experimental results, is now validated through these high-level calculations, which we have added to the revised version of the manuscript and SI.

The manuscript is recommended for publication basically as it is. I have just a few comments and questions that may generate a manuscript minor revision:

- I am not sure I follow the argument about the solvated electron. Is the hypothesis that the light grey peak in the 300 nm spectrum the result of S_0 - S_4 excitation \rightarrow auto-detachment to form a solvated electron \rightarrow photoionization of the solvated electron? Surely, such a process would necessarily be sequential, since the S_4 auto-detaching state would likely have a finite

lifetime, and the entire process could therefore not be directly probed with MP spectroscopy? I can only imagine such a state can be probed, as the authors suggest towards the end of the manuscript, with a time resolved (i.e., sequential) pump-probe experiment. Maybe, as per Occam’s razor, a simpler explanation exists. For instance, have the authors tried to fit the 300 nm photoelectron spectrum using only one Gaussian? Or, other states that were left out from the analysis on the basis of Koopmans’ theorem may have some small contributions as well?

The 300 nm spectrum could be fit with single Gaussians with $eKE_{\text{meas}} = 0.9 \pm 0.1$ eV (Fig. 1, below). If this single feature were attributed to direct non-resonant two-photon detachment from S_0 , it would give an S_0 - D_0 VDE of 7.3 ± 0.2 eV. If it were attributed to resonance-enhanced two-photon detachment via the $n\pi^*$ state at 4.04 eV, it would give an S_0 - D_0 VDE of 7.2 ± 0.2 eV. These values are consistent with our other measurements (Fig. S4) and although it is not possible to rule out the possibility that the spectrum should be fit with a single Gaussian, we believe that the improved fit with two Gaussians and closer agreement with theory for the S_0 - D_0 VDE suggest that the processes giving rise to the spectrum are better described by fitting two Gaussians. We have added Fig. 1 (below) to the SI (Fig. S3) along with a discussion of the two fits. We have also added a reference to Fig. S3 in the first paragraph of the Discussion.

Figure 1: MP detachment photoelectron spectrum (grey line) of 20 μM $p\text{-HBDI}^-$ in 3/2 (v/v) water/methanol recorded following photoexcitation at 300 nm, plotted as a function of eKE and fit with single Gaussians (red lines) [(a)-(c)] and two Gaussians (grey shading and dashed line) [(d) and Fig 2a in the paper]. Single Gaussians have been fit to the high eKE edge (a), low eKE edge (b) and in between (c). The residuals for all the fits are shown in the lower panels. For all single Gaussian fits, $eKE_{\text{meas}} = 0.9 \pm 0.1$ eV, for which $\Delta_{\text{meas}} = -0.06 \pm 0.1$ eV and thus $eKE_{\text{corr}} = 0.96 \pm 0.2$ eV. Assuming direct non-resonant two-photon detachment from S_0 , this would give S_0 - $D_0 = 7.3 \pm 0.2$ eV.

Regarding the timescale for the formation of solvated electrons, it was not our intention to imply that we are certain that we observe detachment from solvated electrons. The timescale of our measurement is defined by the duration of our laser pulses, which is ~ 150 fs. In principle, autodetachment from the S_4 state will be fast because it is an excited-shape resonance with respect to the quasi-continuum of the solvated electron; however, it will only be possible to know the timescale by making a time-resolved measurement, which we plan to do.

We have edited the text at the end of the discussion to make it clearer that we suspect solvated electrons will be observed in a time-resolved experiment and that *if* the processes contributing to the 300 nm photoelectron spectrum are best understood by two Gaussians and *if* the timescale for formation of a solvated electron from S_4 is faster than the timescale of our measurement, the lower eKE Gaussian could be attributed to the formation of a solvated electron (energetically). We have also changed the way we present the low eKE peak in Fig 2a in the manuscript to be a black dashed outline, rather than using solid light grey shading, to emphasise that its assignment is only tentative.

- To be clear, when the authors mention using an “energy difference method” with PBE0,

does that mean calculating the S0 and D0 for the same geometry + environment at the PBE0 level of theory and then finding the energy difference?

This is correct. We have rewritten the last sentence of the first paragraph of the computational part of the Methods section. “The first VDE was calculated as the energy difference between the S₀ and D₀ states in the geometry of the anion at the PBE0/(aug)-cc-pVDZ//EFP level of theory for a series of model systems with increasingly larger size, up to 19,000 water molecules (Fig. S9). All water molecules, except for those included in the quantum mechanical part, were treated as EFPs.”

- The approach used to determine the effect of inelastic scattering (Δ_{meas}) is unclear. It would be good to add more details in the SI. I was particularly confused by Fig. S2. Why is the “used” correction in this study sometimes taken from the water plot, and sometimes in between the water and methanol plots?

For $eKE = 0.5 - 1.2$ eV, we used the water values. There are two values of Δ_{meas} for $eKE = 1.1$ eV, so we took the average of these. There are no water values for $eKE > 1.5$ eV but the methanol values are fairly constant across the range $eKE = 1.1 - 2.1$ eV, so we took an average of the water and methanol values at $eKE = 1.5$ eV. We have added this text to the caption for Fig. S2 in the SI. We have also added the second decimal place (that we used in our determination of VDE) to the values of Δ_{meas} , in both the manuscript (Table 1) and the SI (Table S1).

Please note that we have also corrected a mistake in our propagation of errors. The revised errors in eKE_{meas} and VDE have all decreased by 0.1 eV.

Minor points:

Page 2 top. I think the following change: “macroscopic response” → “macromolecular-level response and, in turn, biological function” is necessary.

We agree and changed the text.

Page 2 bottom. “the most direct way” → “one of the most direct way”

We agree and changed the text.

Page 3 middle. “rapid isomerization” → “ultrafast double-bond isomerization”.

We agree and changed the text.

Reviewer 2

This manuscript describes measurements of photoelectron spectra of the green fluorescent protein (GFP) chromophore in a water-methanol solution using a liquid microjet. The spectra are measured by multiphoton ultraviolet photoelectron spectroscopy. Both resonant and non-resonant pathways are identified and the vertical detachment energies (VDEs) corresponding to the ground and several electronically excited states of both the anion and the neutral molecule are reported. The experimental results are compared with QM/MM calculations relying on density-functional theory, the effective-fragment potential method and a very large number of water molecules to reach converged detachment energies.

This study is very interesting and important because it demonstrates the capability of liquid microjet photoelectron spectroscopy to access detachment (or ionization) energies of protein chromophores, even when they have rather low solubilities. The GFP chromophore is of particular interest for many biochemical applications and is therefore an excellent choice. Overall, the study has been carried out with care, the experimental data recorded at six different wavelengths of the detachment laser are very detailed and the performed non-linear fits are convincing. The theory effort is also particularly impressive, especially given the size and

quality of the performed calculations, the demonstrated convergence of the energies with increasing numbers of water molecules and the agreement with the experimental results.

However, the main result of this work are absolute VDEs and their comparison with theory. The determination of detachment and ionization energies from liquid-microjet photoelectron spectroscopy is an important yet still not fully understood topic that has been the object of many recent publications regarding, in particular, the correct referencing of absolute energies and the effect of inelastic electron scattering.

Specifically, I would like the authors to address the following comments:

1. Inelastic scattering of electrons in water/methanol has been convincingly shown to modify the observed photoelectron spectra in Refs. 33 and 34. The authors have made use of these results to extract a correction " Δ_{meas} " (see Table 1 and Fig. S2). Inelastic scattering can cause energy loss, i.e. a positive Δ_{meas} , but it is unclear how the authors justify the negative values of Δ_{meas} up to measured kinetic energies of 1 eV. This aspect is not discussed in the present work and it appears to be unphysical.

Inelastic scattering of low energy electrons could result in excitation or de-excitation of normal modes of water and a negative Δ_{meas} could arise from the de-exciting inelastic scattering processes. However, we attribute the negative Δ_{meas} to the reduced probability of escape for low eKE electrons that arises from the reduced solid angle required for electrons to hit the surface with a high enough normal velocity component to overcome the barrier for escape (Luckhaus et al., *Sci. Adv.* 3, e1603224, 2017). This reduces the contribution from the lower eKEs and shifts the measured maximum intensity of the photoelectron spectrum to higher eKEs. Accounting for the impact of photoelectron inelastic scattering and the probability of escape on photoelectron spectra is non-trivial. Luckhaus et al. (*Sci. Adv.* 3, e1603224, 2017) have developed scattering calculations to account for them and Nishitani et al (*Sci. Adv.* 5, eaaw6896, 2019) have developed a spectral retrieval method. Both approaches have been employed to determine true photoelectron spectra of the solvated electron in water from series of UV photoelectron spectra and, whilst they agree in terms of the value for the VBE (the peak of the true photoelectron spectrum), there is not a consensus on the shape of the true UV photoelectron spectrum. Therefore, at this stage, we can only account for peak shifts, not photoelectron spectral profiles. Shifting the peaks using the data available in the literature, accounts for shifts that have contributions from both inelastic scattering *and* the escape probability. It is worth noting that we have used this procedure to determine the water $1b_1$ VIE to be 11.4 ± 0.1 eV (Fig. S1). This is in good agreement with the literature value of 11.3 ± 0.1 eV (Kurahashi, *JCP* 140, 174506, 2014) and gives us confidence in our approach.

We have revised the text in the manuscript (Photoelectron spectra, paragraph 2) to explain that our procedure approximates contributions from the escape probability as well as inelastic scattering. We also renamed Section 1.2 in the SI to "Estimation of Δ_{meas} ", to make it clear that this is an estimation, not an absolute determination which is an open question.

2. Recent work on photoelectron spectroscopy (PES) of static liquids has shown that, similar to solid-state PES, the Fermi levels of a conductive solution and the photoelectron analyzer are equilibrated (Tissot et al., *Top. Catal.* 59, 605, 2016). Since the work functions of the solution and the analyzer are different, this situation results in an energetic offset of the vacuum levels, which will lead to incorrect absolute energies unless it is compensated. In solid-state PES, this effect is well known and addressed by biasing the sample. Similar ideas have been applied to liquid microjets by Olivieri et al., *PCCP* 18,

29506, 2016 and by Perry et al., JPC Lett. 11, 1789, 2020, who found significant deviations from ionization energies reported earlier. In the present work, the authors briefly mention the streaming potential, but they do not address the issue of energy referencing, which is important when reporting absolute energies.

We agree that energy referencing is important when reporting absolute energies and it is something that we have taken into account. We do not apply an external voltage to the liquid sample. Recent work by Nishitani et al., JCP 152, 144503, 2020, has shown that although applying an external voltage can flatten the vacuum level potential around the liquid microjet, the entire photoelectron spectrum is also shifted. What we do is measure both the streaming potential and the difference between the work functions of the liquid microjet and the detector, using the procedure we described in Riley et al., Rev. Sci. Instrum. 90, 083104, 2019.

Briefly, we calibrate the TOF using NO in the absence of the liquid microjet with the magnet in two different positions: the optimal magnetic bottle position for sample measurements and with the magnet translated away from the interaction region for streaming potential measurements. For the streaming potential measurements, the eKE measured following 2+1 REMPI of Xe is plotted as a function of distance x between the ionisation point and the liquid-microjet. Measurements are fitted using $eKE(x) = eKE_{\text{field-free}} - L\phi_{\text{str}}/(L+x) + V$ (Tang et al., Chem. Phys. Lett. 494, 111, 2010), where L is the distance between the ionisation point and the skimmer, $eKE_{\text{field-free}}$ is the eKE following 2+1 photoionisation of Xe when the jet is not running, ϕ_{str} is the streaming potential. In Rev. Sci. Instrum. 90, 083104, 2019, we stated that V accounted for additional fields in the magnetic bottle spectrometer with the liquid-microjet nozzle in place. With hindsight, this could have been referred to as the difference between the work functions of the liquid microjet and the detector. We obtain ϕ_{str} from our fit to the streaming potential equation given above. We obtain V from the difference between our photoelectron spectra of Xe with and without the liquid microjet, both recorded with the magnetic bottle geometry set for streaming potential measurements.

Using this procedure, we have determined the VIE of water $1b_1$ to be 11.4 ± 0.1 eV (Fig. S1). This is in good agreement with the literature value of 11.3 ± 0.1 eV (Kurahashi, JCP 140, 174506, 2014) and gives us confidence in our absolute energies.

We have added a section to the SI explaining this procedure (Section 1.6). We have changed the first sentence in the Methods section of the paper to, “Photoelectron spectra were recorded using our liquid-microjet magnetic-bottle photoelectron spectrometer and procedures that have been described in detail elsewhere.” We also changed the last sentence in the Methods section of the paper to, “Photoelectron spectra of NO were recorded to convert TOF to eKE and photoelectron spectra of Xe were recorded to determine the energy resolution ($\Delta E/E \sim 1\%$), streaming potential and difference between the work functions of the liquid microjet and the detector.”

3. The authors used 30 mM NaF to compensate for the effects of the streaming potential. They do not justify this choice, nor do they show that the streaming potential is effectively compensated. This would, however, be an important and straightforward experimental verification that would contribute to support the reported VDEs.

We measure the streaming potential and account for it in our analysis. We have added the results of these measurements for the data presented in this paper to the SI (Fig. S8).

4. The transmission function of photoelectron spectrometers, including magnetic-bottle devices, tends to decrease significantly at very low kinetic energies. This, in turn, will

affect the determination of the VDEs involving the slowest electrons because the corresponding line shapes will not be Gaussian. Have the authors verified the assumed energy independence of the low-energy electron transmission function (e.g. by gas-phase measurements on NO)?

We record two-photon non-resonant MPI photoelectron spectra of NO to determine the eKE to eV calibration for each experiment. Example NO spectra are presented in Fig. 2, together with a plot of the relative areas of individual peaks. Since it would be unphysical for the transmission efficiency for photoelectrons with 0.33 eV eKE to be higher than those with 0.59 or 0.78 eV eKE ($v^+ = 1$ data), we assume that the transmission efficiency is close to 100% for photoelectrons with 0.33 eV eKE. We do not have enough data to plot the eKE dependence of the transmission function at low eKE; however, we have tested the robustness of our fits. We fit the 3 spectra in which there is significant photoelectron signal for eKEs < 0.3 eV (249.7 nm, 430 nm and 439.8 nm) using all the data (Fig. 3, top panels) and compare with fits to data with eKE > 0.2 , 0.3 and 0.4 eV (Fig. 3, lower 3 panels). For the 0.2 and 0.3 eV cut-offs, the fits yield the same values for eKE_{meas} as the fits to all the data (to 1 d.p.). For the 0.4 eV cut-offs, the fits yield values for eKE_{meas} that are within the experimental error of those reported for the fits to all the data. This gives us confidence in the fits we report in the paper.

Figure 2: Left: 1+1 non-resonant photoelectron spectra of NO at 4 wavelengths, normalised on the areas of the peaks corresponding to $v^+ = 0$ (labelled at the top). Right: plot of the relative areas of the peaks in the NO photoelectron spectra. The areas of the peaks are fit with line shapes modelled by Gaussian and exponential functions (Buchner et al., Rev. Sci. Instrum. 81, 113107, 2010) and the plotted error bars correspond to those obtained from the fits. The numbers printed alongside the points are the eKEs of the peak maxima.

We have added a section to the SI discussing the instrument transmission function and robustness of the fit at low eKE in which we include Figs 2 and 3.

Figure 3: MP detachment photoelectron spectra (grey lines) of $20 \mu\text{M } p\text{-HBDI}^-$ in 3/2 (v/v) water/methanol recorded following photoexcitation at (a) 249.7 nm, (b) 430 nm and (c) 439.8 nm, plotted as a function of eKE and fit with Gaussian line shapes (colours match those in Fig. 2 in the manuscript), for a range of low eKE cut-offs. Numbers represent values for $e\text{KE}_{\text{meas}}$ obtained from the fits.

Reviewer 3

This paper presents photoelectron spectrum of the GFP chromophore in a liquid water microjet. The main objective is to determine the vertical detachment energy (VDE) of the chromophore, and this is found to be 6.8 eV, about 4 eV higher than the VDE of the gas phase species. The higher VDE is consistent with the results for many aqueous anions. While the determination of this value is of some interest, the main result of the paper seems pretty straightforward and there is little reason to publish in a high profile journal such as Nature Comm. The authors have done a thorough job in figuring out wavelength dependence of their photoelectron spectra; this complex dependence results from the energetic requirement of two photons to detach an electron of the solvated chromophore. One thus has to sort out the non-resonant vs resonant multiphoton processes. It would have been much easier to do this experiment in a synchrotron where one-photon detachment would be energetically accessible.

It is unlikely that the VDEs could have been measured at a synchrotron using X-ray photoelectron spectroscopy because the chromophore is only sparingly soluble in aqueous solution (20 μM in 3/2 (v/v) water/methanol). Such low solubility would have made it extremely difficult to obtain a good signal-to-noise ratio. Our paper highlights the value of MP UV photoelectron spectroscopy for making such measurements. Moreover, this MP UV photoelectron spectroscopy study of the deprotonated GFP chromophore in aqueous solution is significant because it reveals that the VDE and pattern of higher lying electronically excited states in aqueous solution are similar to those in the protein.

Reviewer #1 (Remarks to the Author):

The authors have addressed all my comments quite clearly. In summary, this is a very interesting manuscript also including state-of-the-art quantum chemical computation of VDE using excitation to a diffuse orbital centered outside of the water solvent box: I find this approach perfectly valid even if it is, in my view, original.

I recommend acceptance of the revised version of the manuscript without further changes.

Reviewer #2 (Remarks to the Author):

The authors have made a commendable effort in revising their manuscript. In my previous report, I have argued that the main results of this work were absolute vertical detachment energies (VDEs) and their comparison with theory. I have pointed out 4 limitations to the quantitative determination of such VDEs from liquid-microjet photoelectron spectroscopy. In response to these points the authors have provided detailed responses and some new data.

Unfortunately, these amendments confirm and even accentuate the concerns that I have expressed, rather than alleviating them. The most significant problem is the determination of the absolute energies themselves. This involves the compensation of the streaming potential and the vacuum-level offset. The authors appear to agree with this assertion and they show in Fig. S8 how they determined these quantities (note that ϕ_{sp} in the insets of Fig. S8 should probably read ϕ_{str} , as in the caption). Although the authors appear to be studying the same sample (solution) and are using the same spectrometer, the values of the streaming potential and the vacuum level offsets that they determined are completely different from measurement to measurement. The streaming potentials vary from -0.043 ± 0.003 eV to $+0.173 \pm 0.013$ eV, i.e. they differ markedly and even change sign. This is not only unexpected, but it really appears to point to some uncontrolled experimental conditions. The same concern also applies to the determined vacuum level offsets, which vary from $V = -0.19$ eV to $V = -0.02$ eV. In this case, the underlying data are not shown and the error bars are not provided. Nevertheless, the considerable variation is again unexpected, and is even hard to justify because the vacuum-level offsets should only depend on the nature and surface properties of the solution and the spectrometers and therefore be reasonably constant from measurement to measurement. In addition to the unexplained variations, the authors will also need to reconsider the equation that they used to determine the streaming potential and the vacuum level offset. For this purpose, they have used the equation $eKE(x) = eKE(\text{field-free}) - L\phi_{str}/(L + x) + V$

and they have assigned to V “with hindsight” the meaning of the vacuum-level offset. Unfortunately, this is physically incorrect, because the effect of a vacuum-level offset alone (i.e. in the absence of a streaming potential) also leads to a shift of the photoelectron kinetic energies that varies as a function of the jet displacement. In any case, the seemingly uncontrolled variation of the determined parameters, combined with their incorrect interpretation seriously questions the reliability of the reported VDEs. The authors then argue that they trust their absolute energies because the value of the 1b1 ionization energy of liquid water that they determined from Fig. S1 (11.4 \pm 0.1 eV) agrees with a previous literature value, for which they quote 11.3 \pm 0.1 eV and refer to Kurahashi et al., JCP 2014. First of all, the reliable determination of the 1b1 binding energy from Fig. S1 is itself questionable because the 1b1 photoelectron band is strongly overlapped with two neighboring contributions, which are assigned to F- and methanol, respectively. The strong overlap itself might already lead to a correlation of the fit parameters, but what is a more serious limitation are the underlying assumptions, i.e. the energy-independent transmission function of the photoelectron spectrometer (which is not fulfilled, see below) and the Gaussian lineshape of all 3 photoelectron bands (which is unlikely to be true, both physically and technically because of the intrinsically asymmetric lineshapes of magnetic-bottle spectrometers). The other problem is the used reference value (of 11.3 \pm 0.1 eV) itself, which was obtained in a paper that exclusively discussed the effect of the streaming potential and did not consider the additional effect of the vacuum-level offset. That paper contains experimental data (cp. Fig. 4 and Fig. 12 in Kurahashi et al, JCP 2014) that cannot be explained in terms of the streaming potential alone, but can be explained if different vacuum-level offsets are being assumed for different spectrometer configurations (which were indeed used). Finally, it is worth commenting on the shift of the photoelectron spectrum by the applied bias voltage, mentioned by the authors with reference to Nishitani et al., JCP 2020. This shift was previously observed and discussed by Perry et al., JPC Lett. 2020 (see their Fig. 3d). The results of Perry et al. are validated by the fact that the measured gas-phase 1b1 ionization energy of 12.65(9) eV is in quantitative agreement with the literature value. In the opinion of this referee, this excludes any possible systematic error in the absolute binding energy reported by Perry et al.

The second problem with the current manuscript is the transmission function of the photoelectron spectrometer at low energies. In their Fig. S6, the authors show the results of measurements of slow photoelectrons obtained from the ionization of NO. They use this data to argue that the transmission function of their spectrometer does not significantly decrease down to kinetic energies of 0.3 eV. While it is unclear what they exactly mean, it is clear that the transmission function does vary substantially between 0.33 eV and 1.09 eV. Specifically, the scatter of the data points at $v=1$ in the right panel of Fig. S6 is a measure for the variation of this transmission function (assuming the Condon principle to be valid for the photoionization of NO), such that I am led to conclude that the transmission function varies by almost a factor of 2 in this energy range. This would imply that the measured line shapes at low energies differ significantly from the physical lineshapes of the photoelectron spectra. Since all results of this work depend on the quantitative determination of overlapping photoelectron spectra at low energies, it is clear that the results can only be trusted if the energy-dependent transmission function is quantitatively known and taken into account in the data analysis.

The third problem is the correction for inelastic electron scattering (Table S1). In my first report I have argued that negative values for this correction are unlikely to be physical. The authors have replied that negative values might arise from deexcitation of vibrational modes in liquid water. This is also unlikely to be correct because a net deexcitation of vibrational modes would require a population inversion of vibrational modes, i.e. a non-thermal population distribution, which is impossible. Alternatively, the cross section for de-excitation would have to be much larger than the cross section for excitation, which would violate the principle of detailed balance.

As a second explanation for the negative energy correction, the authors argue that the reduced probability of electron escape from the liquid surface might be the explanation. In this argument, they appear to have been influenced by Luckhaus et al., *Sci. Adv.* 2017. Besides the fact that Luckhaus et al. have adopted all of their scattering parameters from measurements on amorphous ice, rather than liquid water, and that they have not provided an independent validation of the scattering data, they have also adopted the (unverified) assumption of a 1.0 eV escape barrier from the amorphous ice work. The escape barrier for liquid water is actually much smaller, i.e. it has been determined (from the extrapolation of experimental cluster data) to lie between 0 and 0.12 eV by Coe et al., *JCP* 107, 6023 (1997). Therefore, it still seems questionable how the negative energy corrections arise. They are more likely to arise from some experimental artefacts, related to the different aspects discussed in this report, than to a well-defined physical mechanism.

Overall, I am led to conclude that there are too many unresolved problems with the experimental data, their collection, their analysis and their interpretation, to recommend publication at this stage. Without reliable quantitative results, the relevance of the paper is insufficient for publication in *Nature Communications*.

The authors have made a commendable effort in revising their manuscript. In my previous report, I have argued that the main results of this work were absolute vertical detachment energies (VDEs) and their comparison with theory. I have pointed out 4 limitations to the quantitative determination of such VDEs from liquid-microjet photoelectron spectroscopy. In response to these points the authors have provided detailed responses and some new data.

- Unfortunately, these amendments confirm and even accentuate the concerns that I have expressed, rather than alleviating them. The most significant problem is the determination of the absolute energies themselves. This involves the compensation of the streaming potential and the vacuum-level offset. The authors appear to agree with this assertion and they show in Fig. S8 how they determined these quantities (note that ϕ_{sp} in the insets of Fig. S8 should probably read ϕ_{str} , as in the caption).
- Although the authors appear to be studying the same sample (solution) and are using the same spectrometer, the values of the streaming potential and the vacuum level offsets that they determined are completely different from measurement to measurement. The streaming potentials vary from -0.043 ± 0.003 eV to $+0.173 \pm 0.013$ eV, *i.e.* they differ markedly and even change sign. This is not only unexpected, but it really appears to point to some uncontrolled experimental conditions.
- The same concern also applies to the determined vacuum level offsets, which vary from $V = -0.19$ eV to $V = -0.02$ eV. In this case, the underlying data are not shown and the error bars are not provided. Nevertheless, the considerable variation is again unexpected, and is even hard to justify because the vacuum-level offsets should only depend on the nature and surface properties of the solution and the spectrometers and therefore be reasonably constant from measurement to measurement.
- In addition to the unexplained variations, the authors will also need to reconsider the equation that they used to determine the streaming potential and the vacuum level offset. For this purpose, they have used the equation $eKE(x) = eKE(\text{field-free}) - L\phi_{str}/(L+x) + V$ and they have assigned to V “with hindsight” the meaning of the vacuum-level offset. Unfortunately, this is physically incorrect, because the effect of a vacuum-level offset alone (*i.e.* in the absence of a streaming potential) also leads to a shift of the photoelectron kinetic energies that varies as a function of the jet displacement.

In any case, the seemingly uncontrolled variation of the determined parameters, combined with their incorrect interpretation seriously questions the reliability of the reported VDEs.

We have made a number of improvements to our instrument and procedures. We have coated the nozzle-holder, skimmer and catcher in graphite to flatten the potential in the interaction region. We also now use a catcher instead of a cold-trap. This lowers the background pressure in the interaction region from $\sim 10^{-4}$ mbar to $\sim 2 \times 10^{-5}$ mbar, and thus minimises scattering of low eKE electrons from gas-phase water. It also extends the time for which liquid-microjet measurements can be made so we can record all the data we need in one day under identical conditions. Instead of measuring the value of the streaming potential and accounting for it, we now change the salt solution and flow-rate to minimise the streaming potential.

The variations in streaming potential measurements and work-function differences in our original work were due to the measurements being made on different days, most likely with small differences in the solutions and in the precise configuration in the interaction region. Now that we are able to record all measurements in a single day, this issue is resolved.

Our new procedures have been described in detail in Section S1 of the revised Supplementary Information.

- The authors then argue that they trust their absolute energies because the value of the $1b_1$

ionization energy of liquid water that they determined from Fig. S1 (11.4 ± 0.1 eV) agrees with a previous literature value, for which they quote 11.3 ± 0.1 eV and refer to Kurahashi *et al.*, JCP 2014. First of all, the reliable determination of the $1b_1$ binding energy from Fig. S1 is itself questionable because the $1b_1$ photoelectron band is strongly overlapped with two neighboring contributions, which are assigned to F- and methanol, respectively.

- The strong overlap itself might already lead to a correlation of the fit parameters, but what is a more serious limitation are the underlying assumptions, i.e. the energy-independent transmission function of the photoelectron spectrometer (which is not fulfilled, see below) and the Gaussian lineshape of all 3 photoelectron bands (which is unlikely to be true, both physically and technically because of the intrinsically asymmetric lineshapes of magnetic-bottle spectrometers).
- The other problem is the used reference value (of 11.3 ± 0.1 eV) itself, which was obtained in a paper that exclusively discussed the effect of the streaming potential and did not consider the additional effect of the vacuum-level offset. That paper contains experimental data (cp. Fig. 4 and Fig. 12 in Kurahashi et al, JCP 2014) that cannot be explained in terms of the streaming potential alone, but can be explained if different vacuum-level offsets are being assumed for different spectrometer configurations (which were indeed used).
- Finally, it is worth commenting on the shift of the photoelectron spectrum by the applied bias voltage, mentioned by the authors with reference to Nishitani *et al.*, JCP 2020. This shift was previously observed and discussed by Perry *et al.*, JPC Lett. 2020 (see their Fig. 3d). The results of Perry *et al.* are validated by the fact that the measured gas-phase $1b_1$ ionization energy of $12.65(9)$ eV is in quantitative agreement with the literature value. In the opinion of this referee, this excludes any possible systematic error in the absolute binding energy reported by Perry *et al.*

The precise value of the ionisation energy of water is subject to ongoing discussion. Since the measurement reported by Perry *et al.*, papers from Suzuki's group (Thürmer *et al.* J. Phys. Chem. A 2021, 125, 2492) and a consortium of groups (Thürmer *et al.*, arXiv 2104.01630) have presented values that are 0.3 eV lower than the value reported by Perry *et al.*. As a result of this debate, the ionisation energy of water may not be the best benchmark for our measurements. We believe that the robustness of our revised procedures and the self-consistency of our measurements and agreement with theory are good enough. We have also recorded photoelectron spectra following $1 + 1$ detachment of phenolate (one half of the HBDI⁻ chromophore) at 306 nm (well below the S_1 VEE). The photoelectron spectrum has a perfect Gaussian line profile with a FWHM and VDE that are consistent with an X-ray PES measurement (7.1 ± 0.1 eV; Ghosh *et al.*, J. Phys. Chem. B 2012, 116, 7269). We have not presented the data here or in the Supplementary Information for this paper as we plan to include it in another publication.

The second problem with the current manuscript is the transmission function of the photoelectron spectrometer at low energies. In their Fig. S6, the authors show the results of measurements of slow photoelectrons obtained from the ionization of NO. They use this data to argue that the transmission function of their spectrometer does not significantly decrease down to kinetic energies of 0.3 eV. While it is unclear what they exactly mean, it is clear that the transmission function does vary substantially between 0.33 eV and 1.09 eV. Specifically, the scatter of the data points at $v+1$ in the right panel of Fig. S6 is a measure for the variation of this transmission function (assuming the Condon principle to be valid for the photoionization of NO), such that I am led to conclude that the transmission function varies by almost a factor of 2 in this energy range. This would imply that the measured line shapes at low energies differ significantly from the physical lineshapes of the photoelectron spectra. Since all results of this work depend on the quantitative determination of overlapping photoelectron spectra at low energies, it is clear that the results can only be trusted if the energy-dependent transmission function is quantitatively known and taken into account in the data analysis.

We have now recorded a more extensive set of NO photoelectron spectra that allow us to determine the instrument function more accurately (Fig. S1) and we now account for it in our analysis of our

spectra (Fig. S1 and S2). The transmission efficiency of our spectrometer is around 87% at 0.4 eKE, 76% at 0.3 eKE and 54% at 0.2 eKE. We have investigated the effect of fitting Gaussians to our spectra after cutting off the low eKE components at 0.2 eV, 0.3 eV and 0.4 eV and find that the centres of the fitted Gaussians do not change with the position of the low eKE cutoff (within experimental error). This gives us confidence in our procedure.

The third problem is the correction for inelastic electron scattering (Table S1). In my first report I have argued that negative values for this correction are unlikely to be physical. The authors have replied that negative values might arise from deexcitation of vibrational modes in liquid water. This is also unlikely to be correct because a net deexcitation of vibrational modes would require a population inversion of vibrational modes, i.e. a non-thermal population distribution, which is impossible. Alternatively, the cross section for de-excitation would have to be much larger than the cross section for excitation, which would violate the principle of detailed balance.

As a second explanation for the negative energy correction, the authors argue that the reduced probability of electron escape from the liquid surface might be the explanation. In this argument, they appear to have been influenced by Luckhaus *et al.*, Sci. Adv. 2017. Besides the fact that Luckhaus *et al.* have adopted all of their scattering parameters from measurements on amorphous ice, rather than liquid water, and that they have not provided an independent validation of the scattering data, they have also adopted the (unverified) assumption of a 1.0 eV escape barrier from the amorphous ice work. The escape barrier for liquid water is actually much smaller, i.e. it has been determined (from the extrapolation of experimental cluster data) to lie between 0 and 0.12 eV by Coe *et al.*, JCP 107, 6023 (1997). Therefore, it still seems questionable how the negative energy corrections arise. They are more likely to arise from some experimental artefacts, related to the different aspects discussed in this report, than to a well-defined physical mechanism.

We agree with the reviewer that there are unanswered questions about the accuracy of the scattering cross-sections employed in the simulations in the 2017 Sci. Adv. paper. However, what the experimental measurements in that paper do tell us is that the apparent eBEs shift both above and below the true value, as determined from EUV measurements. Importantly, the spectra are also distorted significantly and the distortion is dependent on photon energy. The fact that we can fit our photoelectron spectra to Gaussian profiles, which is certainly not possible for the data published in the 2017 Sci. Adv. paper, therefore seems at odds with eKE dependent inelastic scattering. This caused us to rethink our analysis of our data.

We have carried out one-dimensional scattering simulations to investigate the effect of inelastic scattering on eKE loss as a function of the depth at which the photoelectrons are emitted. This analysis is presented in Section S2 of the Supplementary Information. In summary, we find that photoelectrons generated with eKE > 1 eV within 5 nm of the surface are unlikely to scatter with solvent and will escape with almost no loss of eKE. Photoelectrons within 5 nm of the surface with lower eKEs (< 1 eV) are more likely to scatter but the loss of eKE is minimal and within the experimental error of our measurements. Therefore, together with the effectiveness of Gaussian fits to the experimental data, this suggests that scattering effects are insignificant. We attribute this to the enhanced surface concentration of weakly soluble organic molecules, such as the the GFP chromophore presented in this work. The Gaussian profile we observe here can be considered analogous to those observed in X-ray photoelectron spectroscopy studies in which the probing depth is known to be only around 1 nm. As a result, we no longer correct for inelastic scattering in our analysis of our data. The improvements to our experimental methods and procedures outlined in our responses above, the self-consistency of our measurements and calculations, and our measurement of aqueous phenolate, give us confidence in the accuracy of our measurements and their analysis.

Overall, I am led to conclude that there are too many unresolved problems with the experimental data, their collection, their analysis and their interpretation, to recommend publication at this stage. Without reliable quantitative results, the relevance of the paper is insufficient for publication in Nature Communications.

To conclude, we would like to thank the reviewer for highlighting the imperfections in our first set of measurements that resulted in us re-recording our data with revised procedures. Our revised measurements of the S_0 - D_0 and S_0 - D_1 VDEs of p -HBDI⁻ are self-consistent and in excellent agreement with theory. Although the overall interpretation of our results has not changed significantly, the values we have determined for the VDEs are slightly different. Whilst we were revising our experimental procedures and recording new photoelectron spectra, we also improved our computational approach to calculating the S_0 - D_0 and S_0 - D_1 VDEs.

The text and figures in the manuscript have been revised to reflect the new data and revised numerical values. The overall interpretation and conclusions have not changed. The Supplementary Information has been completely rewritten to describe our revised procedures and characterisation of the instrument function, validity of the low eKE cutoff and inelastic scattering simulations. We hope that with these substantial revisions, our manuscript will now be considered suitable for publication in Nature Communications.

Reviewers' comments:

Reviewer #2 (Remarks to the Author):

The authors have made an impressive effort to revise and improve their manuscript. They have repeated all of their measurements, improving the control over their experimental conditions. These developments have allowed them to acquire data within a single day of measurements, which appears to have eliminated the uncontrolled variations in the experimental conditions. The new experimental results are therefore clearly more reliable.

The authors have also reconsidered their treatment of electron scattering and have now concluded that inelastic scattering was negligible for the solute they studied. They have performed one-dimensional scattering calculations that appear to support this conclusion. Overall, these additions and improvements are all highly welcome as they remove some of the major problems present in the previous version.

However, what is still missing is a conclusive experimental result showing that the absolute values of the reported VDEs (the main result of the present work) are quantitatively accurate. In contrast to their previous version, the authors are no longer showing the data that contained the water, ethanol and F- signals. While I agree with the authors that the absolute ionization energy of liquid water is the subject of an ongoing controversy, the former data had the advantage that they could at least be compared with experimental literature values (cited by the authors). In the present version of the manuscript, such a comparison is no longer possible. The only possible comparison concerns the VDE of the solvated electron for which the authors determine the value of 3.3 ± 0.2 eV (Table 1 and page 12), which does not agree with the literature value of 3.76 eV, mentioned by the authors. This discrepancy should be discussed, taking into account the known solvation time scales of aqueous electrons. In the same context, the authors mention their measurements of phenolate (VDE = 7.1 ± 0.1 eV), which agrees well with X-ray photoelectron spectroscopy. Unfortunately, the results are not shown and it is moreover difficult to assess the reliability of the reference data (Ghosh et al, J. Phys. Chem. B 2012, 116, 7269), recorded at a time when the intricacies of accurate liquid-jet photoelectron spectroscopy were far from being understood. Furthermore, the authors quote the good agreement of their experimental and calculated VDEs as supporting the self-consistency of their results. This argument is weak, particularly in the logic of the authors, given that the latest review about calculating ionization energies of solvated species (DOI: 10.1002/wcms.1519) concludes that the best theoretical vertical ionization energy of liquid water is 11.6 eV (Table 7), a value that agrees with the results of Perry et al. JPCL 2020, which the present authors appear to question in their Section S1.2 when they state that "although accurate re-calibration after applying the bias is then required to obtain accurate absolute energies".

To conclude, the present version of the manuscript is clearly considerably improved, but it falls short of clearly and unambiguously making the case of reporting the quantitatively accurate VDEs that would merit publication in Nature Communications.

The authors have made an impressive effort to revise and improve their manuscript. They have repeated all of their measurements, improving the control over their experimental conditions. These developments have allowed them to acquire data within a single day of measurements, which appears to have eliminated the uncontrolled variations in the experimental conditions. The new experimental results are therefore clearly more reliable. The authors have also reconsidered their treatment of electron scattering and have now concluded that inelastic scattering was negligible for the solute they studied. They have performed one-dimensional scattering calculations that appear to support this conclusion. Overall, these additions and improvements are all highly welcome as they remove some of the major problems present in the previous version.

These additions and improvements address all the issues highlighted by the reviewer in the previous version apart from the reviewer's concerns about the VIE of water, which we address below.

However, what is still missing is a conclusive experimental result showing that the absolute values of the reported VDEs (the main result of the present work) are quantitatively accurate. In contrast to their previous version, the authors are no longer showing the data that contained the water, ethanol and F- signals. While I agree with the authors that the absolute ionization energy of liquid water is the subject of an ongoing controversy, the former data had the advantage that they could at least be compared with experimental literature values (cited by the authors). In the present version of the manuscript, such a comparison is no longer possible.

As well as the controversy surrounding the absolute value of the ionisation energy of water, since we have now concluded that inelastic scattering is negligible for weakly soluble organic solutes with an enhanced surface concentration, it does not make sense to benchmark our data against a photoelectron spectrum of bulk water for which inelastic scattering would not be at all negligible. However, a photoelectron spectrum of phenolate, which is a building block of HBDI⁻, is appropriate. We chose not to present this in the previous response as we were intending publishing it elsewhere; however, it is now presented and discussed below.

The only possible comparison concerns the VDE of the solvated electron for which the authors determine the value of 3.3 ± 0.2 eV (Table 1 and page 12), which does not agree with the literature value of 3.76 eV, mentioned by the authors. This discrepancy should be discussed, taking into account the known solvation time scales of aqueous electrons.

We discussed the value of 3.3 ± 0.2 eV on p.12 of the manuscript. We interpreted the value as being that of a *partially* solvated electron, by which we mean the early stages of solvation. Aqueous electrons become fully solvated on timescale of 10s of picoseconds. Recent work by Suzuki and co-workers (J. Phys. Chem. B 2019, 123, 3769-3775) gives timescales for the various processes in the formation of a solvated electron: charge transfer to solvent state \rightarrow contact pair state \rightarrow solvent separated state \rightarrow solvated electron \rightarrow electron in conduction band, as around 0.15, 0.4-1.1, 13-32, 300-560 ps, respectively, for tetrabutylammonium iodide (TBAI) which has a hydrophobic tail and therefore higher concentration at the surface (like HBDI⁻ and phenolate). The peaks of the decay associated binding energy spectra attributed to the decay of the initially formed contact pair state are around 0.5 eV lower than those attributed to the decay of the solvated electron (Fig. 5 in J. Phys. Chem. B 2019, 123, 3769-3775). Although the precise value of the shift is concentration and system dependent, and likely to be dependent on the method of formation, we believe that our interpretation of the peak around 3.3 eV corresponding to a partially solvated electron (or contact-pair state) is consistent with this data. We have added a reference to Suzuki's work at the end of the sentence where we state that our VDE is consistent with that of a partially solvated electron (Ref. 37).

In the same context, the authors mention their measurements of phenolate (VDE= 7.1 ± 0.1 eV), which agrees well with X-ray photoelectron spectroscopy. Unfortunately, the results are not shown and it is moreover difficult to assess the reliability of the reference data (Ghosh et al, J. Phys. Chem. B 2012, 116, 7269), recorded at a time when the intricacies of accurate liquid-jet photoelectron spectroscopy were far from being understood.

The photoelectron spectrum following 1 + 1 detachment of phenolate at 306 nm (well below the S_1 VEE, see Fig. 1 below) is presented in Fig. 2 below. The spectrum has a Gaussian line profile with a FWHM of 1.0 eV and maximum at 0.82 ± 0.2 eV eKE, corresponding to a VDE of 7.3 ± 0.2 eV. This is consistent with an X-ray PES measurement (7.1 ± 0.1 eV VDE and ~ 1 eV FWHM, Ghosh et al., J. Phys. Chem. B 2012, 116, 7269). The experimental procedure for recording this spectrum was identical to that described in our Supplementary Information and the flat streaming potential measurement obtained before this measurements is presented in Fig. 3 below.

Whilst we agree that the intricacies of accurate X-ray photoelectron spectroscopy measurements of liquids were not understood completely in 2010, the values obtained are not expected to vary *significantly* from their true values. For example, Fig. 4 in Chem. Sci., 2021, 12, 10558–1058 (cited as DOI: 10.1039/d1sc01908b in our previous response) shows the spread of values obtained for the ionisation energy; apart from a particularly early measurement (2004) and the value reported by Perry et al., the spread of values from measurements with photon energies > 25 eV is < 0.2 eV.

Figure 1: UV-vis absorption spectrum of of 1 mM aqueous solution of phenolate with 5 mM NaOH.

Figure 2: Left: 1 + 1 detachment photoelectron spectrum (black) of 1 mM aqueous solution of phenolate with 5 mM NaOH at 306 nm. The fitted Gaussian (blue) has a peak centre of 0.82 ± 0.2 eV. The conservative error is the same as that for HBDI^- , using the same arguments presented in the Supplementary Information. Right: Residual from the Gaussian fit.

Figure 3: Streaming potential measurement for 1 mM aqueous solution of phenolate with 5 mM NaOH (equivalent to Fig. S4 in the Supplementary Information) obtained before recording the photoelectron spectrum presented in Fig. 2, to confirm that the potential in the interaction region of the photoelectron spectrometer was flat and to determine a value for the vacuum level offset between the interaction region and the spectrometer.

Furthermore, the authors quote the good agreement of their experimental and calculated VDEs as supporting the self-consistency of their results. This argument is weak, particularly in the logic of the authors, given that the latest review about calculating ionization energies of solvated species (DOI: 10.1002/wcms.1519) concludes that the best theoretical vertical ionization energy of liquid water is 11.6 eV (Table 7), a value that agrees with the results of Perry et al. JPCL 2020, which the present authors appear to question in their Section S1.2 when they state that “although accurate re-calibration after applying the bias is then required to obtain accurate absolute energies”.

The review that is cited (DOI: 10.1002/wcms.1519) is about an implicit solvation model. We use a more advanced explicit solvation model, which includes nearest water molecules treated at the same level of theory as the quantum part (thus including all kinds of interactions), as well as all other water molecules treated explicitly through the EFP method. Also, we use one of the most accurate multireference multiconfiguration methods (XMCQDPT2/EFP) to treat the QM part. This method also agrees well with the values obtained at the DFT/EFP level (which is completely different), where the size of the model system can be varied systematically to ensure that the saturation limit is reached with respect to a number of explicit water molecules. These calculations are state-of-the-art, combining both the high accuracy and the very large size of the model system. We are confident that these calculations are trustworthy. Moreover, a calculation of the phenolate anion at the XMCQDPT2/EFP level gives a VDE of 7.0 eV, which is in good agreement with the X-ray experiment and our own measurement.

In terms of the agreement between the VIE of liquid water cited in Table 7 of DOI: 10.1002/wcms.1519 and that measured by Perry et al. JPCL 2020, it should be noted that the latter is not in agreement with X-ray PES measurements from other groups, including the most recent data (see Fig. 4 in Chem. Sci., 2021, 12, 10558–10558).

We have removed the phrase “although accurate re-calibration after applying the bias is then required to obtain accurate absolute energies” from Section 1.2 in the Supplementary Information.

We would like to emphasise that along with the new and important experimental data on HBDI^- , we also introduce a highly accurate way of calculating VDEs. Therefore, we have all reasons to believe that both the experiment and the theory provide a consistent and unified qualitative picture of the lowest lying VDEs in HBDI^- .

REVIEWERS' COMMENTS

Reviewer #2 (Remarks to the Author):

The authors have revised their manuscript one more time. Even after these latest revisions, the present manuscript fails to convince that reliable ionisation energies have been unequivocally determined. The present results indeed still point to, both, uncontrolled variations of the experimental conditions and inconsistent interpretations, which suggest that the authors will not be able to get much further with their present experimental technique. Given that the ionisation energies are the main result of this work, I cannot recommend publication of this work in Nature Communications.

Specifically, in Fig. 3 of their reply, the authors showed a "streaming potential measurement" of a 1 mM aqueous solution of phenolate with 5 mM NaOH, which is offset by 0.04 eV from the reference value. In Fig. S4, the offsets amounted to 0.18 eV before and 0.1 eV after the measurements of p-HBDI⁻. All of these offsets are attributed to the "vacuum level offset between the interaction region and the analyser". First, the large variations of a quantity that should be constant for a given apparatus again suggest that important experimental parameters influencing the experimental results are uncontrolled. Second, as I commented in an earlier report, the vacuum-level offset (between the solution and the analyser) leads to a change of the kinetic energy of the electrons that varies (logarithmically) with the distance from the liquid jet. Therefore, the observed (constant) offset can actually not originate from a vacuum-level offset. Moreover, typical vacuum-level offsets between dilute aqueous salt solutions and analysers are on the order of 0.4-0.6 eV, as demonstrated by Olivieri et al., PCCP 2016, Perry et al., JPCL 2020 and Nishitani et al., JCP 2020. These numbers do not match with the interpretation of the present experimental results, casting significant doubts on the reliability of the determined ionisation energies.

Overall, the present work thus represents an interesting attempt to access the ionisation energy of solvated p-HBDI⁻, but the results unfortunately remain doubtful. I can therefore not recommend publication in Nature Communications. Since the present results could be of interest to the liquid-microjet photoelectron community, I recommend publication in a more specialized journal, such as Communications Chemistry.

Reviewer #4 (Remarks to the Author):

The paper from Fielding and coworkers reports new and fundamentally interesting results by using an emerging spectroscopic technique applied to the chromophore at the heart of the much-used

green fluorescent protein. The manuscript version I was provided is a carefully detailed work that I find satisfying in its level of technical detail. Important issues are addressed, and the authors go to pains not to sweep complicated issues under the rug.

Liquid jet photoelectron spectroscopy promises a new approach to unravel complex excited state chromophore dynamics (particularly the outcome of excitation to higher electronic states) while including the complexity of aqueous/biochemical environment. Liquid jet photoemission is an emerging technique in the sense that some of the “operating parameters” are still being hashed out by the ten or so groups worldwide most active in its development. As the reviewer/author exchange for this manuscript makes clear, there remains technical disagreement between the various groups on how best to achieve highest reproducibility and accuracy for the vertical detachment energies that are reported by the method.

These are reasonable disagreements based on the different backgrounds and perspectives of the different researchers involved. Several papers have recently appeared addressing the various experimental issues (streaming potential, vacuum level offset, transmission function, suitable liquid ionization potentials that can serve as a reference). The debate is healthy, and this paper serves a valuable purpose in furthering the debate, even if community convergence is still to be achieved.

I do find the review comments of referee 2 in the first two rounds of review/rebuttal both reasonable and constructive; similarly, I find the response of the authors to be serious and note that they have been willing to pour considerable new effort into making best attempt to resolve. The fact that, as a result of the second round of review comments, a complete experimental re-investigation was carried out is significant - these experiments are difficult and the graduate student responsible for the original measurements had no doubt moved on since initial submission. For more detailed thoughts on reviewer’s specific concerns, I address the reader to a longer section below.

Overall, I find the exchange to be technically illuminating and helpful to the scientific process, particularly if the review/rebuttal appears with the published article. I do regard the author’s efforts and rebuttal to be sufficient to justify publication at this stage. While the numerical vertical detachment energies are important, they are not the sole justification for this report to be published (in other words, I disagree with the statement “without reliable quantitative results, the relevance of the paper is insufficient...”). This contribution is an important demonstration applying a new technique to a biologically relevant system, and therefore moves the liquid jet resonant photoemission technique from the prototype-only stage to one where an open biophysical question is being addressed. It is important to stress that most of the issues being argued about introduce quantitative errors of the order of 0.1 – 0.2 eV (similar to the expected precision of theory being compared to). Moreover, as pointed out by the authors, these corrections do not change the interpretation, assignment, and conclusions. So, while further robust debate on these various experimental complications is likely to continue, this does not preclude the usefulness in showcasing

a new technique to a problem of broad interest to chemists, theorists, biochemists and structural biologists.

Author's response to specific issues raised by reviewer 2 in the second response:

By performing new experiments, the authors have resolved the day-to-day variations in streaming potentials and work-function differences, which was one of the largest repeated critiques from the reviewer. In addition, the authors have yielded on the transmission function by performing a very nice and non-trivial set of measurements to characterize this. Most other liquid jet groups have not taken this step to characterize their instruments as carefully.

Then the reviewer's concern expressed in their second set of bullets are strongly based on the reviewer's perception that the use of 11.3 eV for the water 1b1 energy reference (as has been done by several groups hitherto) is flawed. They reach this point of view based on the work of Perry et al, which places this reference at 11.7 eV based on a different methodology. While Perry's paper is important, there is a more recent paper (Thurmer et al, Chem. Science (2021) 12, 10558–1058) specifically expanding on this issue. By using yet another methodology and squarely addressing the work-function concerns of the reviewer, the Thurmer paper reaffirms the 11.3 eV value. While many in the community do not regard the issue of the water reference ionization potential as fully resolved, I do agree with the authors that focusing on the ionization energy of water as a reference is providing a distraction from the main results of this work. I note that it is, however, that it was somewhat less satisfactory that the authors reassured themselves on this point via data with a different reference system (aqueous phenolate) that they did not initially wish to share. I am glad that this now forms part of the review record.

The final (and somewhat separate issue in my mind) to be resolved is the physical (rather than experimental) effect of inelastic scattering on the kinetic energy of the measured photoelectrons. This topic is also being intensely debated in the community. Practical steps are being taken to resolve this, based on applying electron/water scattering simulations. The authors provide (in section S2 of the SI) a simplified modeling of the inelastic scattering effect albeit still relying on ice cross sections that reviewer 2 finds objectionable but are still the only ones available. (More detail on how the modeling calculations presented were performed would have been nice). Increasingly detailed work on inelastic scattering is in progress by other groups, but the authors current presentation is helpful and may be sufficient for this system because of the interfacial activity of the GFP chromophore.

Reviewer 2

The authors have revised their manuscript one more time. Even after these latest revisions, the present manuscript fails to convince that reliable ionisation energies have been unequivocally determined. The present results indeed still point to, both, uncontrolled variations of the experimental conditions and inconsistent interpretations, which suggest that the authors will not be able to get much further with their present experimental technique. Given that the ionisation energies are the main result of this work, I cannot recommend publication of this work in Nature Communications.

We disagree with the comment that there are “uncontrolled variations of experimental conditions” and address the specific criticisms relating to it below. It is worth emphasising that, in addition to determining HB DI^- VDEs, which are significant measurements in their own right, we also introduce a general procedure for determining VDEs of organic molecules in aqueous solution using liquid-microjet photoelectron spectroscopy that is as accurate as current understanding of the technique allows. To clarify this in the main text, after “to determine the first three vertical detachment energies (VDEs)” we have added “to an accuracy of ± 0.2 eV, representing uncertainties in the effect of inelastic scattering of low energy electrons (< 5 eV) and the vacuum level offset (Supplementary Methods).”

Specifically, in Fig. 3 of their reply, the authors showed a “streaming potential measurement” of a 1 mM aqueous solution of phenolate with 5 mM NaOH, which is offset by 0.04 eV from the reference value. In Fig. S4, the offsets amounted to 0.18 eV before and 0.1 eV after the measurements of p-HB DI^- . All of these offsets are attributed to the “vacuum level offset between the interaction region and the analyser”. First, the large variations of a quantity that should be constant for a given apparatus again suggest that important experimental parameters influencing the experimental results are uncontrolled.

We find, consistently, that for liquid-microjet measurements recorded over a period of up to 8 hours, the vacuum level offset increases by up to 0.1 eV. We suspect that this is attributed to increased water vapour in the interaction region or the catcher, or other changes in experimental conditions that occur during an experimental run, as stated in the supplementary methods. This shift is less than our overall experimental uncertainty, ± 0.2 eV. It is worth noting that other groups have not tended to report streaming potential measurements before and after liquid-microjet photoelectron spectroscopy measurements.

Second, as I commented in an earlier report, the vacuum-level offset (between the solution and the analyser) leads to a change of the kinetic energy of the electrons that varies (logarithmically) with the distance from the liquid jet. Therefore, the observed (constant) offset can actually not originate from a vacuum-level offset.

The vacuum level is flat across the interaction region, as stated in the supplementary methods. We coat the magnet and skimmer with graphite and adjust the salt concentration in the liquid to align the vacuum level of the liquid with that of graphite (e.g. Fig. 1B in Nishitani et al., J. Chem. Phys. 152, 144503, 2020). The vacuum level offset to which we refer is between the skimmer and detector and is, therefore, independent of the distance between the ionisation point and the liquid jet.

Moreover, typical vacuum-level offsets between dilute aqueous salt solutions and analysers are on the order of 0.4-0.6 eV, as demonstrated by Olivieri et al., PCCP 2016, Perry et al., JPCL 2020 and Nishitani et al., JCP 2020. These numbers do not match with the interpretation of the present experimental results, causing significant doubts on the reliability of the determined ionisation energies.

We do not have a definitive explanation for why our vacuum level offset is lower than those of these three groups who have measured theirs. We have all coated the key components of our interaction regions in graphite; however, there will be differences depending on whether the inner surface of the analyser is coated with graphite and on whether bias voltages are applied to the flight tube or to the liquid-microjet to flatten the potential in the interaction region. The inside of our analyser is not coated with graphite and we have not applied bias voltages to either the flight tube or the liquid jet.

Overall, the present work thus represents an interesting attempt to access the ionisation energy of solvated p- HBDI⁻, but the results unfortunately remain doubtful. I can therefore not recommend publication in Nature Communications. Since the present results could be of interest to the liquid-microjet photoelectron community, I recommend publication in a more specialized journal, such as Communications Chemistry.

Reviewer 4

The paper from Fielding and coworkers reports new and fundamentally interesting results by using an emerging spectroscopic technique applied to the chromophore at the heart of the much-used green fluorescent protein. The manuscript version I was provided is a carefully detailed work that I find satisfying in its level of technical detail. Important issues are addressed, and the authors go to pains not to sweep complicated issues under the rug.

Liquid jet photoelectron spectroscopy promises a new approach to unravel complex excited state chromophore dynamics (particularly the outcome of excitation to higher electronic states) while including the complexity of aqueous/biochemical environment. Liquid jet photoemission is an emerging technique in the sense that some of the “operating parameters” are still being hashed out by the ten or so groups worldwide most active in its development. As the reviewer/author exchange for this manuscript makes clear, there remains technical disagreement between the various groups on how best to achieve highest reproducibility and accuracy for the vertical detachment energies that are reported by the method.

These are reasonable disagreements based on the different backgrounds and perspectives of the different researchers involved. Several papers have recently appeared addressing the various experimental issues (streaming potential, vacuum level offset, transmission function, suitable liquid ionization potentials that can serve as a reference). The debate is healthy, and this paper serves a valuable purpose in furthering the debate, even if community convergence is still to be achieved.

I do find the review comments of referee 2 in the first two rounds of review/rebuttal both reasonable and constructive; similarly, I find the response of the authors to be serious and note that they have been willing to pour considerable new effort into making best attempt to resolve. The fact that, as a result of the second round of review comments, a complete experimental re-investigation was carried out is significant - these experiments are difficult and the graduate student responsible for the original measurements had no doubt moved on since initial submission. For more detailed thoughts on reviewer’s specific concerns, I address the reader to a longer section below.

Overall, I find the exchange to be technically illuminating and helpful to the scientific process, particularly if the review/rebuttal appears with the published article. I do regard the author’s efforts and rebuttal to be sufficient to justify publication at this stage. While the numerical vertical detachment energies are important, they are not the sole justification for this report to be published (in other words, I disagree with the statement “without reliable quantitative results, the relevance of the paper is insufficient...”). This contribution is an important demonstration applying a new technique to a biologically relevant system, and therefore moves the liquid jet resonant photoemission technique from the prototype-only stage to one where an open biophysical question is being addressed. It is important to stress that most of the issues being argued about introduce quantitative errors of the order of 0.1 – 0.2 eV (similar to the expected precision of theory being compared to). Moreover, as pointed out by the authors, these corrections do not change the interpretation, assignment, and conclusions. So, while further robust debate on these various experimental complications is likely to continue, this does not preclude the usefulness in showcasing a new technique to a problem of broad interest to chemists, theorists, biochemists and structural biologists.

Author’s response to specific issues raised by reviewer 2 in the second response: By performing new experiments, the authors have resolved the day-to-day variations in streaming potentials and work-function differences, which was one of the largest repeated critiques from the reviewer. In addition, the authors have yielded on the transmission function by performing a very nice and non-trivial set of measurements to characterize this. Most other liquid jet groups have not taken this step to characterize

their instruments as carefully.

Then the reviewer's concern expressed in their second set of bullets are strongly based on the reviewer's perception that the use of 11.3 eV for the water 1b1 energy reference (as has been done by several groups hitherto) is flawed. They reach this point of view based on the work of Perry et al, which places this reference at 11.7 eV based on a different methodology. While Perry's paper is important, there is a more recent paper (Thurmer et al, Chem. Science (2021) 12, 10558–1058) specifically expanding on this issue. By using yet another methodology and squarely addressing the work-function concerns of the reviewer, the Thurmer paper reaffirms the 11.3 eV value. While many in the community do not regard the issue of the water reference ionization potential as fully resolved, I do agree with the authors that focusing on the ionization energy of water as a reference is providing a distraction from the main results of this work. I note that it is, however, that it was somewhat less satisfactory that the authors reassured themselves on this point via data with a different reference system (aqueous phenolate) that they did not initially wish to share. I am glad that this now forms part of the review record.

The final (and somewhat separate issue in my mind) to be resolved is the physical (rather than experimental) effect of inelastic scattering on the kinetic energy of the measured photoelectrons. This topic is also being intensely debated in the community. Practical steps are being taken to resolve this, based on applying electron/water scattering simulations. The authors provide (in section S2 of the SI) a simplified modeling of the inelastic scattering effect albeit still relying on ice cross sections that reviewer 2 finds objectionable but are still the only ones available. (More detail on how the modeling calculations presented were performed would have been nice). Increasingly detailed work on inelastic scattering is in progress by other groups, but the authors current presentation is helpful and may be sufficient for this system because of the interfacial activity of the GFP chromophore.

We believe that all the details and references to cross-sections required to reproduce our simple one-dimensional Monte-Carlo simulations are already provided in the supplementary methods. We are in the process of developing more comprehensive simulations that will be described in more detail in a future publication and it is our plan to make the more comprehensive code available, for those who may wish to use it.